# *Mycobacterium* Transcriptional Factor BlaI Regulates Cell Division and Growth and Potentiates β-Lactam Antibiotic Efficacy Against *Mycobacteria*

**DOI:** 10.3390/microorganisms13102245

**Published:** 2025-09-25

**Authors:** Junqi Xu, Mingjun Zhang, Fuling Xie, Junfeng Zhen, Yuerigu Abuliken, Chaoyun Gao, Yongdong Dai, Zhiyong Jiang, Peibo Li, Jianping Xie

**Affiliations:** 1Institute of Modern Biopharmaceuticals, School of Life Sciences, Southwest University, Chongqing 400715, China; 19946951045@163.com (J.X.); zhangmingjun0123@126.com (M.Z.); xiefuling9564@sinovac.com (F.X.); zjf1993zhen@163.com (J.Z.); y14799889695@163.com (Y.A.); gaochaoyun18@163.com (C.G.); yongdong.dai@majorbio.com (Y.D.); jiangzhiyong24@mails.ucas.ac.cn (Z.J.); 2Chongqing Public Health Medical Center, Chongqing 400036, China

**Keywords:** *Mycobacterium tuberculosis*, cell division, BlaI, FtsQ, antibiotic susceptibility

## Abstract

Cell division is critical for the survival, growth, pathogenesis, and antibiotic susceptibility of *Mycobacterium tuberculosis* (Mtb). However, the regulatory networks governing the transcription of genes involved in cell growth and division in Mtb remain poorly understood. This study aimed to investigate the impact of BlaI overexpression on cell division and growth in Mtb and elucidate the underlying mechanisms. *Mycobacterium smegmatis* mc^2^155 was used as the model organism. Recombinant strains overexpressing BlaI were constructed. Scanning electron microscopy (SEM), transmission electron microscopy (TEM), ethidium bromide and Nile red uptake assays, minimum inhibitory concentration (MIC) determination, drug resistance analysis, quantitative real-time PCR (qRT-PCR) assays, and electrophoretic mobility shift assay (EMSA) were employed to assess changes in bacterial morphology, cell wall permeability, antibiotic susceptibility, gene transcription levels, and the interaction between BlaI and its target genes. Overexpression of BlaI disrupted bacterial division in *M. smegmatis*, leading to growth delay, cell elongation, and formation of multi-septa. It also altered the lipid permeability of the cell wall and enhanced the sensitivity of *M. smegmatis* to β-lactam antibiotics. BlaI overexpression affected the transcription of cell division-related genes, particularly downregulating *ftsQ*. Additionally, BlaI negatively regulated the transcription of *Rv1303*—a gene co-transcribed with ATP synthase-encoding genes—inhibiting ATP synthesis. This impaired the phosphorylation of division complex proteins, ultimately affecting cell division and cell wall synthesis. Overexpression of BlaI in Mtb interferes with bacterial division, slows growth, and alters gene expression. Our findings identify a novel role for BlaI in regulating mycobacterial cell division and β-lactam susceptibility, providing a foundation for future mechanistic studies in *M. tuberculosis*, with validation required to assess relevance to clinical tuberculosis—though validation in *M. tuberculosis* and preclinical models is required.

## 1. Introduction

Cell division is a fundamental process in all organisms, requiring precise coordination of multiple proteins. Mycobacteria, characterized by a complex cell wall composed of peptidoglycan, arabinogalactan, and mycolic acid, exhibit unique bipolar growth and division patterns, generating daughter cells of varying lengths. Their intricate cell wall structure necessitates tight regulation of the cell division complex to ensure proper coordination [1]. Proper regulation of cell division is crucial for bacterial survival and pathogenicity, as it enables bacterial growth and reproduction. Asymmetric elongation and division in mycobacteria contribute to antibiotic resistance by generating heterogeneous populations with varying susceptibility levels and promoting the formation of persister cells [2]. For instance, isoniazid—a first-line anti-tuberculosis drug—disrupts cell wall integrity by targeting InhA, an enzyme critical for mycolic acid synthesis [3]. These mechanistic insights may inform future studies on mycobacterial drug responses, provided they are validated in pathogenic strains and in vivo systems.

Proteins involved in cell division, including those associated with peptidoglycan synthesis, are common targets of antibiotics. Examples include FtsZ (Filamenting temperature-sensitive mutant Z), SepF (Septum-forming protein F), FtsA (Filamenting temperature-sensitive mutant A), FipA (FtsZ-interacting protein A), FtsQ (Filamenting temperature-sensitive mutant Q), FtsW (Filamenting temperature-sensitive mutant W), FtsI (Filamenting temperature-sensitive mutant I), BlaR1 (β-Lactamase Regulator 1), BlaI (β-Lactamase Inhibitor), and BlaC (β-Lactamase C), all of which play critical roles in bacterial cell division and peptidoglycan synthesis. During cell division, FtsZ, SepF, and FtsA form the “Z ring,” which constricts the cell membrane and acts as a scaffold for peptidoglycan remodeling enzymes [1,4,5]. Altered FtsZ expression can lead to abnormal cell elongation [6]. Targeting peptidoglycan synthesis is crucial for inhibiting bacterial cell division and maintaining structural integrity, which can be leveraged to enhance the efficacy of anti-tuberculosis drugs. Conversely, β-lactamases contribute to antibiotic resistance by degrading antibiotics that target penicillin-binding proteins involved in peptidoglycan synthesis [7]. In bacteria such as *Staphylococcus aureus*, BlaR1 senses β-lactam antibiotics and triggers the expression of β-lactamase-encoding genes like *blaZ* [8]. In Mtb, β-lactam resistance is attributed to non-classical transpeptidases and the BlaC β-lactamase [9]. BlaI regulates *blaC* expression [10]; understanding these processes is vital for developing strategies to combat bacterial infections and antibiotic resistance.

Mtb *Rv1846c* (BlaI) is a conserved dimeric repressor protein with DNA-binding and dimerization domains [10], highly conserved across *Mycobacterium*, *Staphylococcus aureus*, and *Bacillus licheniformis* [10,11,12,13]. *Mycobacterium smegmatis* is a well-established model for mycobacterial research due to its rapid growth, non-pathogenicity, and conserved cell division machinery with *M. tuberculosis*. Its genetic tractability allowed us to efficiently dissect BlaI’s mechanism, with plans to validate key findings in *M. tuberculosis* subsequently. Our study revealed that BlaI overexpression in *M. smegmatis* led to growth delay, increased bacterial length, and formation of multiple septa. BlaI downregulated the transcription of the cell division-related gene *ftsQ* and the *Rv1303* gene—co-transcribed with ATP synthase-encoding genes. Additionally, BlaI overexpression enhanced *M. smegmatis* sensitivity to β-lactam antibiotics. The overexpression approach was selected for two primary reasons: (1) To definitively establish BlaI’s biological functions (e.g., its influence on cell division), validating its potential as a therapeutic target; (2) To amplify phenotypic outputs, enabling systematic dissection of downstream regulatory networks. While studies on BlaI inhibition (which may modulate growth) could provide complementary insights, this work focuses on its positive regulatory potential to inform mechanistic understanding of mycobacterial physiology, with potential implications for future therapeutic research. These findings in *M. smegmatis* warrant future validation in *M. tuberculosis*, where BlaI’s role in cell division and antibiotic susceptibility can be directly tested.

Notably, *M. smegmatis* differs from *M. tuberculosis* in key aspects including pathogenicity-associated pathways, cell division regulatory networks, and β-lactam stress responses, which limits direct extrapolation of findings to clinical tuberculosis. This study focuses on mechanistic dissection in *M. smegmatis*, with future validation in *M. tuberculosis* planned to address biological relevance.

## 2. Materials and Methods

### 2.1. Strains, Medium, and Growth Conditions

*Mycobacterium smegmatis* mc^2^155 (ATCC 700084) was cultured at 37 °C in Middlebrook 7H9 broth (BD Difco, Franklin Lakes, NJ, USA, Cat. 271310) supplemented with 0.2% (*v*/*v*) glycerol (Sigma-Aldrich, St. Louis, MO, USA, Cat. G5516), 0.05% (*v*/*v*) Tween-80 (Sigma-Aldrich, St. Louis, MO, USA, Cat. P1379), and 50 μg/mL hygromycin (Sigma-Aldrich, St. Louis, MO, USA, Cat. H0637) when required. For solid culture, Middlebrook 7H10 agar (BD Difco, Franklin Lakes, NJ, USA, Cat. 262710) was prepared with the same supplements. Bacterial cultures were grown in 250-mL Erlenmeyer flasks containing 50 mL broth, incubated at 37 °C with shaking at 180 rpm. When the optical density at 600 nm (OD_600_) reached 0.4, acetamide (ACE; Sigma-Aldrich, St. Louis, MO, USA, Cat. A8054) was added to a final concentration of 0.2% (*w*/*v*) for induction, followed by 8 h of incubation under the same conditions. Bacteria were harvested by centrifugation at 8000× *g* for 10 min, washed twice with sterile 1× PBS (pH 7.4), and resuspended in 7H9 broth to a uniform OD_600_ for subsequent experiments.

### 2.2. Construction of Rv1846c Recombinant Strains and Western Blot

The *Rv1846c* gene (ORF length 429 bp; GenBank accession: NC_000962.3) was amplified from *M. tuberculosis* H37Rv genomic DNA using PCR with gene-specific primers:For pALACE vector: Forward 5′-TCAACCGGGATCCGATGAACGAACA-3′ (BamHI site underlined) and Reverse 5′-GCGGACCCATCGATAGTCTCCCTCA-3′ (ClaI site underlined).For pET-28a(+) vector: Forward 5′-CGCAGGAGATCATATGACAATGGCC-3′ (NdeI site underlined) and Reverse 5′-GCGGACACGCCAAGCTTCTCCCTCA-3′ (HindIII site underlined).

Primers were synthesized by Beijing Genomics Institute (Shenzhen, China) and purified by PAGE. PCR amplification was performed using Phusion High-Fidelity DNA Polymerase (NEB, Ipswich, MA, USA, Cat. M0530) with the following cycling conditions: 98 °C for 30 s, 35 cycles of 98 °C for 10 s, 58 °C for 30 s, 72 °C for 30 s, and a final extension at 72 °C for 5 min. Amplified fragments and vectors were digested with corresponding restriction endonucleases (NEB: BamHI, Cat. R0136; ClaI, Cat. R0197; NdeI, Cat. R0111; HindIII, Cat. R0104) at 37 °C for 2 h, then purified using the Omega Gel Extraction Kit (Omega, Norcross, GA, USA, Cat. D2500-02). Ligation was performed with T4 DNA Ligase (NEB, Cat. M0202) at 22 °C for 1 h, and recombinant plasmids were transformed into *E. coli* DH5α (TransGen, Beijing, China, Cat. CD201) for propagation.

Recombinant pALACE-*Rv1846c* was transformed into *M. smegmatis* mc^2^155 by electroporation (2.5 kV, 25 μF, 1000 Ω) using 0.2-cm cuvettes (Bio-Rad, Hercules, CA, USA), followed by recovery in 7H9 broth (without antibiotics) at 37 °C for 2 h. Transformants were selected on 7H10 agar containing 50 μg/mL hygromycin. For protein expression validation, *M. smegmatis* cultures (OD_600_ = 0.8–1.0) were harvested and lysed by sonication (300 W, 3 s on/5 s off, 10 min) in ice-cold 1× PBS, and 100 μg of total protein was separated by 12% SDS-PAGE. Western blot analysis was performed using anti-Myc monoclonal antibody (TIANGEN, Beijing, China, Cat. MA1020) at 1:1000 dilution, followed by HRP-conjugated anti-mouse IgG (TIANGEN, Cat. SA00001-1) at 1:5000 dilution. Signals were detected using ECL Western blotting Substrate (Millipore, Burlington, MA, USA, Cat. WBKLS0500) and exposed to X-ray film for 30–60 s.

### 2.3. CRISPR Interference (CRISPRi)-Mediated Repression of Rv1846c

To generate the *Rv1846c* (ortholog: *MSMEG_3630* in *Mycobacterium smegmatis*) knockdown strain, we utilized a CRISPRi system as previously described. Briefly, the CRISPRi plasmid backbone (Addgene, Watertown, MA, USA, plasmid 166886) was digested with BsmBI-v2 (NEB, Ipswich, MA, USA, R0739L) and gel-purified. Two complementary oligonucleotides encoding a single-guide RNA (sgRNA) sequence targeting the non-template strand of the *MSMEG_3630* open reading frame (ORF) were designed (3630sgup:GGGAGGTACCTGAATTACCAGTCGGCG; 3630sgdw:AAACGCCGACTGGTAATTCAGGTACC), annealed, and ligated into the BsmBI-digested plasmid backbone using T4 DNA ligase (NEB, Ipswich, MA, USA, M0202M). Successful cloning of the *MSMEG_3630*-targeting sgRNA cassette was confirmed by Sanger sequencing.

For transformation, electrocompetent *M. smegmatis* cells were prepared as follows: *M. smegmatis* cultures were grown in 7H9 medium to an OD_600_ of 0.8–1.0, harvested by centrifugation (4000× *g* for 10 min), and washed three times with sterile 10% glycerol. The washed cells were resuspended in 10% glycerol to a final volume equivalent to 5% of the original culture volume. For each transformation, 100 ng of the *MSMEG_3630*-targeting CRISPRi plasmid was mixed with 100 μL of electrocompetent *M. smegmatis* and transferred to a 0.2-mm electroporation cuvette (Bio-Rad, Hercules, CA, USA, 1652086). Electroporation was performed using the Gene Pulser Xcell electroporation system (Bio-Rad, Hercules, CA, USA, 1652660) set at 2500 V, 700 Ω, and 25 μF. Following electroporation, bacteria were recovered in 7H9 medium for 24 h at 37 °C, then plated on 7H10 agar supplemented with hygromycin (to select for CRISPRi plasmid integration) to obtain *Rv1846c* (*MSMEG_3630*)-knockdown transformants.

#### Validation of Knockdown Efficiency

Single colonies of CRISPRi-positive transformants (Ms_pALACE_*Rv1846c*-KD) and empty plasmid controls (Ms_pALACE) were inoculated into 7H9 broth (supplemented with 50 μg/mL hygromycin) and cultured at 37 °C with shaking (180 rpm) to log phase (OD_600_ = 0.8). Total RNA was extracted from 1 mL cultures using Trizol reagent (Invitrogen, Waltham, MA, USA, Cat. 15596026) and purified with the Omega RNA Purification Kit (Norcross, GA, USA, Cat. R6834-02) to remove genomic DNA; RNA concentration/purity was assessed via Nanodrop 2000 (Thermo Fisher, Waltham, MA, USA) (A_260_/A_280_ = 1.8–2.0). A total of 1 μg purified RNA was reverse-transcribed into cDNA using the PrimeScript RT Reagent Kit (Takara, Kusatsu, Japan, Cat. RR037A) with random hexamers. qRT-PCR was performed on a StepOnePlus Real-Time PCR System (Thermo Fisher) using Takara SYBR Premix Ex Taq™ II (Cat. RR820A): the 20 μL reaction system contained 10 μL SYBR Premix, 0.4 μL each of *MSMEG_3630*-specific primers (10 μM; 3630up: TTGGCCTATACGACGATCATGACGG, 3630down: CAGAACCCGCTGCTTGCTCTCGAGT), 2 μL diluted cDNA (1:5), and 7.2 μL nuclease-free water. The housekeeping gene *sigA* served as the internal reference (primers from Section 2.8), with cycling conditions: 95 °C for 30 s, 40 cycles of 95 °C for 5 s/60 °C for 30 s, and a melting curve analysis to confirm product specificity. Relative *MSMEG_3630* transcription was calculated via the 2^−ΔΔCt^ method, with 3 biological replicates (each with 3 technical replicates); statistical significance was determined by unpaired two-tailed *t*-test. As shown in Appendix A, *MSMEG_3630* transcription in Ms_pALACE_*Rv1846c* -KD was reduced by 62.3% ± 5.7% compared to Ms_pALACE (*p* < 0.01), confirming effective CRISPRi-mediated repression of the BlaI ortholog in *M. smegmatis*.

### 2.4. Bacteria Sample Preparation for Scanning Electron Microscope (SEM) and Transmission Electron Microscope (TEM)

Bacterial cultures were centrifuged at 8000× *g* for 10 min, and pellets were fixed in 2.5% glutaraldehyde (Sigma-Aldrich, Cat. G5882) in 0.1 M PBS (pH 7.4) at 4 °C for 3 h, then washed three times with 1× PBS (15 min each).

For SEM: Fixed cells were dehydrated through a graded ethanol series (50%, 70%, 80%, 90%, 100%; 15 min per step), then transferred to a 1:1 mixture of ethanol and tert-butanol (Sigma-Aldrich, Cat. 471780) for 30 min. Samples were snap-frozen at −20 °C for 2 h, freeze-dried (Labconco FreeZone 2.5, Kansas City, MO, USA), sputter-coated with 10 nm gold (Quorum Q150R ES, Lewes, UK), and imaged using a Hitachi SU8010 SEM (Tokyo, Japan) at 3.0 kV acceleration voltage, 3.6–4.0 mm working distance, and 10,000× magnification.

For TEM: Fixed cells were post-fixed in 1% osmium tetroxide (Sigma-Aldrich, Cat. Os101) at room temperature for 2 h, washed with 1× PBS, and dehydrated through graded ethanol (50% to 100%) and acetone (Sigma-Aldrich, Cat. 179124) (15 min per step). Samples were embedded in epoxy resin (Sigma-Aldrich, Cat. 45345), sectioned into 70-nm ultrathin slices using a Leica EM UC7 ultramicrotome, stained with 2% uranyl acetate and lead citrate, and imaged using a Tecnai G2 Spirit TEM at 80 kV.

Image analysis: Cell length was measured using Image J 1.53e software by manually tracing ≥ 200 cells per sample (across three biological replicates). Septa were defined as electron-dense structures ≥ 0.5 μm in width, spanning ≥50% of the cell diameter. Two independent observers blinded to sample groups analyzed images, with inter-observer consistency verified by a Kappa coefficient > 0.85.

### 2.5. Ethidium Bromide and Nile Red Uptake Assays

Bacteria were grown to OD_600_ = 0.8, harvested, washed twice with 1× PBS, and resuspended in 7H9 broth to OD_600_ = 0.8. An amount of 200 μL of bacterial suspension was mixed with 1 μg/mL ethidium bromide (EB; Sigma-Aldrich, Cat. E1510) or Nile red (NR; Sigma-Aldrich, Cat. 72485) in black 96-well plates with clear bottoms (Corning, Corning, NY, USA, Cat. 3603). Fluorescence was measured every 5 min for 60 min using a Tecan Spark 10M microplate reader (Männedorf, Switzerland, EB: excitation 520 nm, emission 605 nm; NR: excitation 543 nm, emission 590 nm). Plates were shaken for 10 s before each reading.

### 2.6. Minimum Inhibitory Concentration (MIC) Determination

Bacterial cultures were grown overnight, washed with 1× PBS, and adjusted to OD_600_ = 0.8. Antibiotics (penicillin, Sigma-Aldrich, Cat. P3032; ampicillin, Solarbio, Beijing, China, Cat. A8180; isoniazid, Sigma-Aldrich, Cat. I3377; rifampicin, Sigma-Aldrich, Cat. R3501; vancomycin, Sigma-Aldrich, Cat. V1130) were serially diluted 2-fold in 96-well plates (Corning, Cat. 3599) to final concentrations ranging from 0.125 to 256 μg/mL. Bacteria were inoculated at 5 × 10^5^ CFU/mL per well and incubated at 37 °C for 24 h. MIC was defined as the lowest antibiotic concentration with OD_600_ < 0.1. Experiments were performed in three biological replicates, each with two technical replicates.

### 2.7. Drug Resistance Analysis of Recombinant Mycobacterium smegmatis

Acetamide-induced wild-type, Ms_*Rv1846c* (*Rv1846c* -overexpressing strains), and Ms_pALACE (empty plasmid strains, as negative controls) were cultured and adjusted to a uniform concentration (OD_600_ = 0.5). Cultures were diluted 10^−5^-fold and inoculated on 7H10 plates containing varying concentrations of ampicillin. Plates were incubated at 37 °C for 3–4 days, and bacterial colonies were counted. Controls without antibiotics were included. Sterile test tubes containing 7H9 liquid medium and ampicillin (two-fold serial dilutions) were prepared, inoculated with 1% bacterial culture, and incubated with shaking at 37 °C. The MIC of each antibiotic against recombinant and control bacteria was determined. Bacteria were then inoculated into medium containing antibiotics ranging from 0 to 16× MIC (1% inoculum) and incubated at 37 °C for 4 h. Cultures were plated on 7H10 plates, colonies were counted after 3 days, and survival rates were calculated. Experiments were repeated three times.

### 2.8. Quantitative Real-Time PCR (qRT-PCR) Assays

Total RNA was extracted using Trizol reagent (Invitrogen, Cat. 15596026) from 1 mL bacterial cultures (OD_600_ = 0.8), followed by purification with the Omega RNA Purification Kit (Cat. R6834-02). RNA concentration was measured using a Nanodrop 2000 (Thermo Fisher), and 1 μg RNA was reverse-transcribed into cDNA using the PrimeScript RT Reagent Kit (Takara, Cat. RR037A). qRT-PCR was performed with Takara SYBR Premix Ex Taq™ II (Cat. RR820A) on a StepOnePlus Real-Time PCR System (Thermo Fisher). The reaction system (20 μL) included 10 μL SYBR Premix, 0.4 μL each primer (10 μM), 2 μL cDNA, and 7.2 μL ddH_2_O. Cycling conditions: 95 °C for 30 s, 40 cycles of 95 °C for 5 s and 60 °C for 30 s, followed by a melting curve analysis (95 °C for 15 s, 60 °C for 1 min, 95 °C for 15 s). Relative gene expression was normalized to *sigA* using the 2^−ΔΔCt^ method. Primers used are listed in Appendix A. Each gene was analyzed in three biological replicates, each with three technical replicates.

### 2.9. Electrophoretic Mobility Shift Assay (EMSA)

Recombinant BlaI protein was expressed in *E. coli* BL21 (DE3) harboring pET-28a(+)-*Rv1846c* by induction with 0.5 mM IPTG (Sigma-Aldrich, Cat. I6758) at 37 °C for 4 h. Cells were lysed by sonication in lysis buffer (50 mM Tris-HCl pH 8.0, 300 mM NaCl, 10 mM imidazole) and purified using Ni-NTA agarose (Qiagen, Hilden, Germany, Cat. 30210) under native conditions. The protein was eluted with buffer containing 250 mM imidazole, dialyzed against storage buffer (50 mM Tris-HCl pH 8.0, 150 mM NaCl, 10% glycerol), and purity was verified by 12% SDS-PAGE.

Promoter regions of *ftsQ* (Rv2151c) and *Rv1303* were amplified from *M. tuberculosis* H37Rv genomic DNA using specific primers (Rv2151c up:GCGTTGACGATCCTGATCTG, Rv2151c down:CAGCTCGTCGTTGATGTTGA: Rv1303 up:ACGACGCTGCTGAAGAAGAT, Rv1303 down:TCGATGCGCTTGATGTTCTG) and purified as described above. Binding reactions (10 μL) contained 200 ng promoter probe, 0–200 ng BlaI protein, 2 μL EMSA buffer (10 mM Tris-HCl pH 8.0, 50 mM NaCl, 1 mM DTT, 5% glycerol), and ddH_2_O. Reactions were incubated at 37 °C for 30 min, then resolved on 6% non-denaturing PAGE (acrylamide:bisacrylamide = 29:1) in 0.5× TBE at 4 °C and 100 V for 45 min. Gels were stained with Gold View™ (Solarbio, Cat. G1020) for 15 min and imaged using a GelDoc XR+ system (Bio-Rad).

### 2.10. ATP Measurement

Intracellular ATP was measured using the Beyotime ATP Assay Kit (Haimen, China, Cat. S0026). Log-phase bacteria (OD_600_ = 0.8) were harvested, resuspended in ATP Lysis Buffer, and lysed by bead-beating (0.1 mm zirconia beads, 3 × 1 min at 60% power, 4 °C; BioSpec, Bartlesville, OK, USA, Cat. 11079101z). Lysates were centrifuged at 12,000× *g* for 5 min at 4 °C, and supernatants were collected. 100 μL diluted ATP Test Solution was mixed with 20 μL sample/standard in black 96-well plates, and luminescence (RLU) was measured immediately using a GloMax luminometer (Promega, Madison, WI, USA). ATP levels were normalized to total protein concentration (BCA Protein Assay Kit, Thermo Fisher, Cat. 23225).

### 2.11. Data Processing and Statistical Analysis

GraphPad Prism 9, Image J, and Microsoft Excel 2017 were used for data statistics and graphing; Adobe Photoshop CC 2020 and Microsoft PowerPoint 2019 were used for image and document processing. Statistical significance was analyzed using *t*-tests and two-way analysis of variance (ANOVA). Sample sizes for all experiments: 3 biological replicates (independent cultures) with 2–3 technical replicates per biological replicate, unless stated otherwise. Normality of data was confirmed using the Shapiro–Wilk test, and homogeneity of variance using the F-test, justifying the use of parametric tests (*t*-tests, ANOVA). For multiple comparisons (e.g., qRT-PCR of multiple genes), the Benjamini–Hochberg false discovery rate (FDR) correction was applied to adjust *p*-values. Statistical power analysis indicated ≥ 80% power to detect a 1.5-fold change in gene expression or cell length, based on pilot data (*n* = 3).

## 3. Results

### 3.1. BlaI Overexpression in M. smegmatis Results in Elongated Cells, Multi-Septum Formation, Altered Cell Wall Permeability, and Increased Susceptibility to β-Lactam Antibiotics

To investigate the role of BlaI (*Rv1846c*), we examined its impact on mycobacterial growth. After constructing recombinant strains, BlaI expression was confirmed by Western blot analysis (Figure 1A). Growth curves revealed a 12-h growth delay in the BlaI-overexpressing strain compared to controls, with growth levels equalizing by 56 h (Figure 1B). Given that bacterial growth delay can affect cell elongation, we compared strain lengths and found that the average length of the Ms_*Rv1846c* strain was significantly increased by SEM (Figure 2A). The average 2.5 μm longer length of Ms_*Rv1846c* may explain the initial growth delay (Figure 2B). Elongated cell length may be associated with abnormal division; consistent with this, TEM revealed that the Ms_*Rv1846c* strain contained cells with multi-septa (Figure 2C). Overexpression of BlaI significantly increased the proportion of cells with bi- or multi-septa by 5% (Figure 2D). *MSMEG_3630*, a BlaI ortholog in *M. smegmatis*, shares 83% amino acid identity with Mtb BlaI. However, His-tag Western blot confirmed Mtb BlaI overexpression (Figure 1A), and phenotypic changes (elongated cells, multi-septa; Figure 2C) were specific to BlaI overexpression, indicating that observed effects were driven by Mtb BlaI rather than endogenous *MSMEG_3630*. Collectively, BlaI overexpression induces multi-septum formation and cell elongation, suggesting a key role in regulating cell division.

Altered cell division function may affect mycobacterial cell wall properties. To explore BlaI’s effects on cell division and cell wall, we assessed cell wall permeability. Ethidium bromide (EB) exhibits weak fluorescence in culture medium but strong fluorescence when bound to intracellular DNA, with limited extrusion from cells. Nile red (NR) is a lipophilic fluorescent dye with weak fluorescence in aqueous solutions but strong fluorescence in non-polar environments (e.g., cytoplasm), widely used to stain intracellular lipids [14]. Fluorescence quantification showed no difference in EB accumulation between Ms_*Rv1846c* and controls (Figure 3A), but NR uptake was significantly higher in Ms_*Rv1846c* than in Ms_pALACE (Figure 3B), indicating that BlaI overexpression alters the lipid permeability of the *M. smegmatis* cell wall.

Given that BlaI overexpression may affect cell division, we hypothesized it could alter *M. smegmatis* responses to cell wall-targeting antibiotics. MIC comparisons between recombinant Ms_*Rv1846c* and Ms_pALACE showed that the MICs of penicillin and ampicillin against Ms_*Rv1846c* were 1/8 of those against Ms_pALACE, with no differences observed for other cell wall-targeting antibiotics (Figure 4A). To further confirm sensitivity to β-lactam antibiotics, strains were treated with gradient concentrations of ampicillin: Ms_*Rv1846c* survival rates decreased gradually with increasing ampicillin concentration and were significantly lower than those of Ms_pALACE (Figure 4B). These results indicate that BlaI overexpression enhances *M. smegmatis* sensitivity to β-lactam antibiotics, likely by altering cell division and cell wall properties.

### 3.2. BlaI Overexpression in M. smegmatis Affects Transcription of Cell Division-Related Genes, Specifically Repressing ftsQ

As a repressor, BlaI may regulate the transcription of target genes. To explore the mechanism underlying BlaI-mediated cell division disruption, we quantified the transcription levels of cell division-related genes (Table 1) in recombinant Ms_*Rv1846c* and control strains using qRT-PCR. BlaI overexpression significantly downregulated *ftsZ*, *ftsB*, *ftsQ*, and *murC*, while upregulating *ftsI*, *murF*, and *chiZ* (Figure 5A–C). Transcription of serine/threonine protein kinase-encoding genes *pknA/B* remained unchanged (Figure 5D). These results indicate that BlaI overexpression alters the transcription of cell division-related genes, thereby disrupting bacterial division.

To investigate how BlaI—a transcriptional factor—regulates target genes, we expressed and purified BlaI protein from recombinant *Escherichia coli* BL21 (Figure 6A). As a known repressor, BlaI may impact mycobacterial cell wall and length by downregulating multiple cell division-related genes. Our study identified FtsQ as a potential direct target of BlaI: EMSA confirmed that BlaI directly binds to the *ftsQ* (Rv2151c) promoter (Figure 6B). Additionally, analysis of GEO data on antibiotic-treated Mtb revealed opposing transcription trends between *blaI* and *ftsQ* (Figure 6C), indicating that BlaI’s negative regulation of *ftsQ* transcription persists under antibiotic exposure. Time-course experiments with Ms_GFP (non-specific protein control) showed that BlaI rapidly and specifically downregulated *ftsQ* from 4 h post-induction, with associated morphological changes. ATP assays confirmed that BlaI overexpression reduced intracellular ATP levels by 42% compared to wild-type, validating direct regulatory effects on cell division and energy metabolism (Appendix A). Time-course analysis showed that *ftsQ* transcription was downregulated by 4 h post-induction, with ATP levels decreasing by 30% at 4 h (Appendix A), supporting a temporal link between BlaI-mediated regulation, ATP depletion, and morphological changes.

### 3.3. BlaI Negatively Regulates Transcription of Rv1303, a Gene Co-Transcribed with ATP Synthase-Encoding Genes

ATP is essential for peptidoglycan biosynthesis and cell division. The *Rv1303* gene was predicted as a BlaI target via the online database (http://tbdb.bu.edu/, accessed on 3 May 2021). *Rv1303* is part of the polycistronic operon *Rv1303-atpBEFHAGDC-Rv1312*, which encodes ATP synthase for ATP production [10]. To verify this, we first used qRT-PCR to detect the transcription level of *MSMEG_3630*—the *M. smegmatis* homolog of *Rv1303*—in Ms_*Rv1846c* and Ms_pALACE. BlaI significantly downregulated *MSMEG_3630* transcription (Figure 7A). EMSA further showed that with increasing BlaI concentration, the migration rate of the BlaI-*Rv1303* promoter complex gradually decreased, and free probes diminished, indicating that BlaI binds to the *Rv1303* promoter region (Figure 7B). Combined with qRT-PCR results, these data demonstrate that BlaI negatively regulates the transcription of *Rv1303*, a gene co-transcribed with ATP synthase-encoding operons.

Promoter sequence analysis of *Rv1303* and *Rv2151c* using the MEME suite (https://meme-suite.org/meme/tools/meme, accessed on 18 January 2022) identified a consensus motif (Figure 7C). EMSA confirmed that BlaI binds to both fragments (Figure 7D,E), suggesting this motif is critical for BlaI-DNA binding.

We also analyzed *Rv1846c* (blaI) transcriptional dynamics under stress conditions (BDQ, INH, starvation) linked to ATP metabolism and cell division [15,16,17]. Results in Appendix A show its expression is dynamically regulated (Log2 fold change: −1.5 to 1.5), with notable trends: near-significant upregulation under BDQ (*p* = 0.061), downward trend under INH, and changes under starvation, aligning with its roles in related pathways (The results of the multi-transcriptome analysis of *Rv1846c* were obtained from www.clipme.top:11725/, accessed on 2 March 2022). Despite partial statistical non-significance, directional consistency supports its endogenous involvement in these processes, reinforcing the physiological relevance of our experimental findings.

To validate the specificity of BlaI’s regulatory roles in cell division, β-lactam susceptibility, and ATP metabolism—while avoiding potential artifacts from overexpression systems—we performed CRISPR interference (CRISPRi)-mediated repression of *Rv1846c* in *M. smegmatis* to generate a loss-of-function model. Scanning Electron Microscopy (SEM) images showed no obvious difference in external cell morphology between the *Rv1846c*-knockdown strain (Ms_pALACE_*Rv1846c*-KD) and the wild-type control (Ms_pALACE), with no signs of the cell elongation observed in the overexpression strain (Figure 8A). Quantitative analysis further confirmed that the average cell length of Ms.Δ*Rv1846c* was comparable to that of Ms_pALACE (~2.1 μm vs. ~2.0 μm, *p* > 0.05), ruling out the possibility of non-specific morphological changes and confirming that BlaI overexpression-induced cell elongation is a direct consequence of BlaI gain-of-function (Figure 8B).

Transmission Electron Microscopy (TEM) analysis of septal formation revealed that Ms.Δ*Rv1846c* exhibited a septum frequency (~18% multi-septate cells) nearly identical to that of Ms_pALACE (~17%), in stark contrast to the ~23% multi-septate cells in the BlaI-overexpressing strain (Figure 8C,D). This result directly demonstrates that BlaI repression reverses the abnormal septal phenotype caused by BlaI overexpression, further validating BlaI’s specific role in regulating mycobacterial cell division.

Given BlaI’s established role as a repressor of blaC (β-lactamase-encoding gene) in Mtb, we next assessed β-lactam susceptibility of the *Rv1846c*-knockdown strain. Ampicillin killing curves showed that Ms_pALACE_*Rv1846c*-KD exhibited drastically enhanced resistance to β-lactam antibiotics: at 8× MIC of ampicillin, the survival rate of Ms.Δ*Rv1846c* was ~72%, whereas that of Ms_pALACE was only ~15% (Figure 8E). This phenotype is the inverse of the BlaI-overexpressing strain (which showed increased sensitivity) and is likely attributed to de-repression of blaC—consistent with BlaI’s canonical function as a blaC repressor. When BlaI is knocked down, its inhibitory effect on blaC is relieved, leading to higher β-lactamase activity and thus enhanced antibiotic resistance.

Finally, to clarify the link between BlaI and ATP metabolism, we quantified intracellular ATP levels in Ms.Δ*Rv1846c*. Unlike the ~42% reduction in ATP observed in the BlaI-overexpressing strain, the knockdown strain showed no significant increase in ATP production compared to Ms_pALACE_*Rv1846c*-KD (102.3 ± 5.7 nmol/mg protein vs. 98.6 ± 4.9 nmol/mg protein, *p* > 0.05; Figure 8F). This result explains why Ms_pALACE_*Rv1846c*-KD did not exhibit shortened cell length (a phenotype that might be expected if ATP were elevated): ATP synthesis in mycobacteria is a multi-factorial process regulated by multiple operons and metabolic pathways, and BlaI-mediated repression of Rv1303 (and its associated ATP synthase operon) is only one of many contributing factors. Thus, while BlaI overexpression can suppress ATP production, its knockdown is insufficient to drive a detectable increase in ATP levels—further supporting the specificity of BlaI’s regulatory role in ATP metabolism (rather than a non-specific global effect).

## 4. Discussion

The complex cell wall of mycobacteria contains peptidoglycan, arabinogalactan, and mycolic acid. During cell division, mycobacteria first form a peptidoglycan-containing septum, then generate two daughter cells via cell wall constriction [14]. This process requires coordinated interactions between multiple proteins to sustain cell division [18]. Alterations in cell wall components or membrane integrity can disrupt normal mycobacterial division, affecting host environment adaptation and antibiotic resistance [19].

Our findings demonstrate that BlaI overexpression leads to delayed bacterial growth, increased cell length, and formation of multi-septate cells (Figure 2), indicating a disruptive effect on bacterial division. Altered bacterial growth patterns are often associated with changes in key cell division genes or proteins, as observed in *Brevibacterium lactofermentum* (DivIVA overexpression) and mycobacteria with modified divisome components [20]. qRT-PCR analysis revealed significant changes in the expression of cell division-related genes (e.g., *ftsZ*, *ftsQ*, *ftsI*) in *Rv1846c*-expressing recombinant strains compared to controls [5,21], supporting our hypothesis that BlaI overexpression affects transcription or translation of key genes/proteins during bacterial division. These findings establish a clear link between BlaI overexpression and disrupted bacterial division, shedding light on underlying mechanisms and identifying BlaI as a regulatory node worthy of further investigation in pathogenic mycobacteria.

In *M. smegmatis*, BlaI overexpression disrupted the coordination of cell division complexes, leading to the formation of a second septum before the first septum completed division. FtsQ, recruited by FtsZ, acts as a critical “late-stage” divisome protein; its interactions with other proteins are essential for peptidoglycan hydrolysis and divisome stabilization [22]. qRT-PCR showed downregulated *ftsQ* expression (Figure 5A), and EMSA confirmed BlaI’s regulatory effect on *ftsQ* (Figure 6B), indicating that BlaI negatively regulates FtsQ expression and disrupts coordination between cell division complexes. While EMSA demonstrates in vitro binding, in vivo functionality requires further validation. These findings provide novel insights into the regulatory mechanisms of *Mtb* cell division, highlighting the inhibitory impact of BlaI on FtsQ expression.

Additionally, BlaI negatively regulates *Rv1303*—a gene co-transcribed with ATP synthase-encoding operons—thereby inhibiting bacterial ATP synthesis. ATP is required for peptidoglycan synthesis during cell division; inhibition of ATP synthesis can cause cell wall damage [10]. Phosphorylation of divisome proteins such as FtsZ and Wag31 is critical for regulating their functions [23,24]. Reduced ATP synthesis decreases phosphate availability, impairing the phosphorylation and functional activity of FtsZ and Wag31. Moreover, essential mycobacterial serine/threonine protein kinases (PknA and PknB) and the sole phosphatase (PstP) play key roles in regulating cell division and cell wall synthesis [25,26,27]. PknA-mediated phosphorylation of FtsZ modulates its GTPase activity [24]. Beyond FtsZ, FipA, and Wag31 (known PknA targets), high-throughput phosphoproteomics have identified phosphorylation sites on other cell division proteins (e.g., FtsI, FtsK, FtsQ) [28,29,30]. This highlights the importance of divisome protein phosphorylation for bacterial cell division. BlaI-mediated inhibition of ATP synthesis is hypothesized to impair divisome protein phosphorylation (based on literature linking ATP levels to phosphorylation [24,28]), but direct evidence for this step remains to be confirmed, ultimately inhibiting cell division (Figure 9). BlaI overexpression enhances mycobacterial sensitivity to β-lactam antibiotics (Figure 4), suggesting that BlaI’s regulatory role in cell division and β-lactam susceptibility identifies it as a candidate for future preclinical investigation, but therapeutic targeting remains speculative without validation in *M. tuberculosis* and identification of specific modulators. While no specific BlaI activators are currently known, our study establishes a mechanistic framework for future drug discovery by defining BlaI’s regulatory roles in key processes: cell division (*ftsZ*) and ATP synthesis (*Rv1303*). The regulatory model proposed in Figure 9 is supported by our experimental data on BlaI-mediated repression of *Rv1303*, reduced ATP levels, and consequent cell division defects. However, the link between ATP depletion and impaired phosphorylation of division proteins remains a hypothesis derived from existing literature; future phosphoproteomic studies are needed to confirm this mechanistic detail.

BlaI’s dual regulatory roles—repression of *blaC* and modulation of cell division/energy metabolism genes—raise questions about their biological relevance and evolutionary logic. Our findings do not contradict BlaI’s established role as a *blaC* repressor [10] but extend its regulon to include cell division (*ftsQ*) and ATP synthesis (*Rv1303*) genes. This multifunctionality reflects a common evolutionary strategy among bacterial regulatory factors (e.g., CRP, which coordinates metabolic and virulence pathways) to adapt to dynamic environmental stressors [31].

For mycobacteria, this layered regulatory mechanism likely serves as a hierarchical response to β-lactam pressure: Under acute, low-level β-lactam exposure, BlaI primarily represses *blaC*, directly reducing β-lactamase-mediated antibiotic degradation—a rapid, energy-efficient first line of defense. Under prolonged or high-concentration antibiotic stress, BlaI’s secondary regulation of *ftsQ* and *Rv1303* becomes prominent: by inhibiting cell division and ATP synthesis pauses bacterial growth, reduces production of antibiotic targets (e.g., peptidoglycan), and facilitates transition to a dormant state until stress diminishes. The direct association between reduced ATP and altered phosphorylation of division proteins remains untested and requires future validation. Pharmacological activation of BlaI is speculative at this stage; our work identifies BlaI as a candidate target, but screening for BlaI modulators and testing their efficacy requires dedicated future studies. Future research should also investigate BlaI expression patterns in clinical *M. tuberculosis* isolates, particularly drug-resistant strains, to assess clinical relevance.

This duality avoids redundancy by addressing distinct stress phases. For a pathogen like *M. tuberculosis*, which encounters diverse host environments, such flexibility enhances survival. Importantly, while *M. tuberculosis* would not naturally overexpress BlaI (as this impairs fitness by disrupting division and energy metabolism), this vulnerability makes BlaI a promising therapeutic target. Pharmacological activation of BlaI could synergistically exploit both its *blaC*-repression and division-arresting activities, amplifying β-lactam efficacy beyond the effects of *blaC* inhibition alone. This study uses *M. smegmatis* as a model to uncover BlaI’s regulatory functions, but its relevance to *M. tuberculosis*, clinical isolates, and in vivo contexts must be determined with further testing. The evolutionary significance of BlaI’s regulation of cell division and antibiotic sensitivity remains unclear. While increased β-lactam sensitivity in *M. smegmatis* may seem counterintuitive for a pathogen, it could reflect a stress response strategy: under high antibiotic pressure, temporary division arrest and proposed ATP depletion may promote dormancy, enhancing survival until stress abates. Pharmacological modulation of BlaI is a theoretical future direction, dependent on follow-up studies in *M. tuberculosis* and preclinical models.

BlaI’s known role in repressing blaC (β-lactamase) [10,28] and our finding that it regulates cell division/ATP synthesis may seem contradictory, but this duality reflects a conserved stress response strategy in bacteria: transcriptional regulators coordinate multiple pathways to optimize survival under fluctuating pressures. Like *E. coli*’s CRP, which balances metabolism and virulence [29], BlaI operates via stress-dependent hierarchy: Under low/acute β-lactam stress, it primarily represses blaC to limit antibiotic degradation—an energy-efficient, rapid resistance mechanism [10]. Under high/prolonged stress, it shifts to regulating ftsQ (cell division) and Rv1303 (ATP synthesis), pausing replication and reducing peptidoglycan (β-lactam targets) to promote dormancy, enhancing long-term survival. Thus, BlaI’s dual functions are complementary: short-term resistance via blaC and long-term survival via division arrest—critical for navigating host environments.

Our study builds on BlaI’s established role in modulating β-lactam resistance while uncovering deeper regulatory nodes, strengthening its potential as a target for combating tuberculosis drug resistance.

## 5. Conclusions

Overexpression of BlaI in *Mycobacterium smegmatis* disrupts cell division, slows growth, and alters β-lactam susceptibility by regulating *ftsQ* and *Rv1303*-associated ATP synthesis. These findings identify a novel regulatory role of BlaI in mycobacterial physiology, but their relevance to *M. tuberculosis* and potential clinical implications require further validation. This work provides a foundation for mechanistic studies in pathogenic mycobacteria.

## Figures and Tables

**Figure 1 microorganisms-13-02245-f001:**
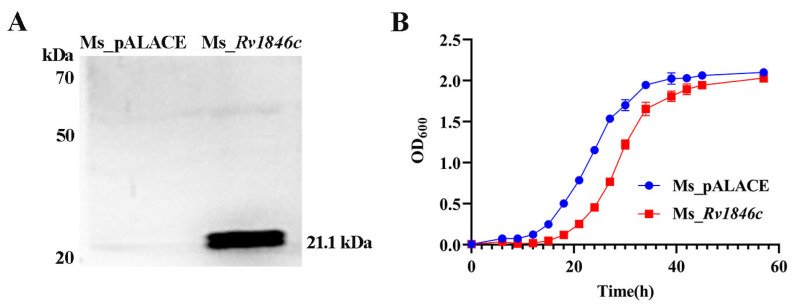
Construction of recombinant *M. smegmatis* Ms_*Rv1846c* and growth curves of Ms_pALACE and Ms_*Rv1846c*. (**A**) Western blot analysis showed a ~21.1 kDa band in Ms_*Rv1846c*, consistent with the expected molecular weight of the BlaI fusion protein (18.5 kDa BlaI + 2.6 kDa His tag), confirming recombinant BlaI expression; Ms_pALACE showed no such band, serving as a negative control. (**B**) Growth curves of Ms_pALACE and Ms_*Rv1846c*. Curves represent three independent biological replicates, each with three technical replicates. Data are shown as mean ± SD. Statistical significance was analyzed using two-way ANOVA.

**Figure 2 microorganisms-13-02245-f002:**
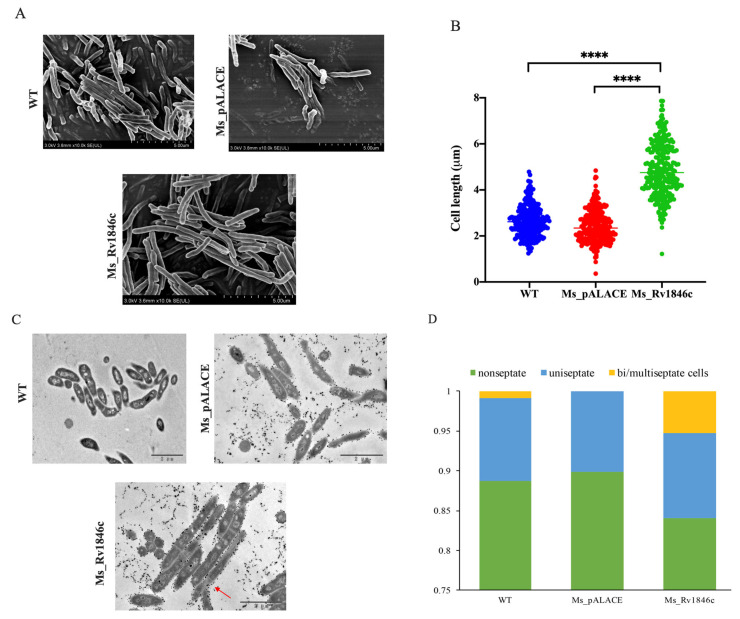
BlaI overexpression increases average cell length and induces a multi-septate phenotype in *M. smegmatis*. (**A**) SEM images of Ms_pALACE and Ms_*Rv1846c* at 10,000× magnification. (**B**) Cell length measurements (266 cells per sample across three biological replicates, total 798 cells). Data were analyzed using one-way ANOVA with Tukey’s post hoc test and plotted as mean ± SD. **** *p* < 0.0001. Lengths were measured using Image J and plotted as a scatter plot using GraphPad Prism 9. Scale bars, 5 μm. (**C**) Representative TEM images of Ms_pALACE and Ms_*Rv1846c*. Red arrows mark bi/multi-septate cells. Scale bars, 2 μm. (**D**) Percentage of non-septate, uniseptate, and bi/multi-septate cells in samples from (**C**). Statistical analysis was performed using *t*-test. Image analysis was conducted blindly by two observers with consistent results (Kappa = 0.89). Representative images from three biological replicates are shown.

**Figure 3 microorganisms-13-02245-f003:**
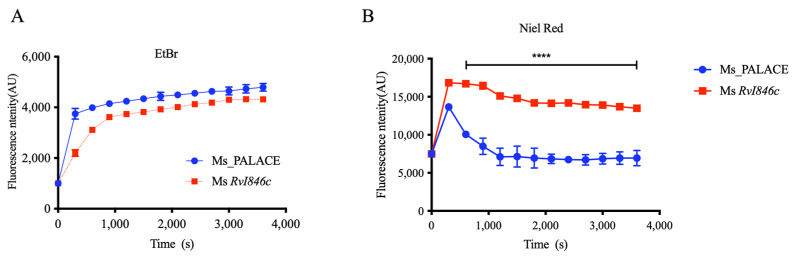
BlaI overexpression alters cell wall properties in *M. smegmatis*. (**A**) Ethidium bromide accumulation in Ms_pALACE and Ms_*Rv1846c*. (**B**) Nile red accumulation in Ms_pALACE and Ms_*Rv1846c*. **** *p* < 0.0001

**Figure 4 microorganisms-13-02245-f004:**
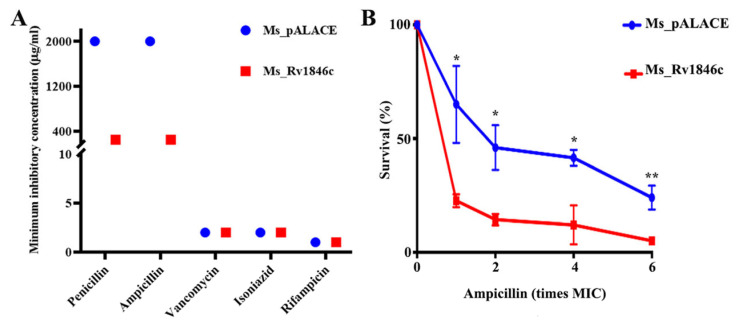
BlaI overexpression enhances *M. smegmatis* sensitivity to β-lactam antibiotics. (**A**) MICs of various antibiotics against *M. smegmatis* strains. (**B**) Survival rates of Ms_pALACE and Ms_*Rv1846c* after 12-h ampicillin treatment. MIC determination and survival assays were performed with three biological replicates. Survival rate data are presented as mean ± SD, with significance analyzed by *t*-test. * *p* < 0.05; ** *p* < 0.01.

**Figure 5 microorganisms-13-02245-f005:**
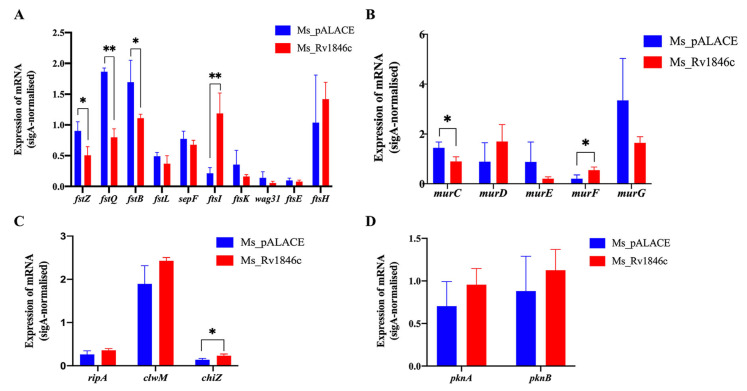
BlaI overexpression alters transcription of cell division-related genes. qRT-PCR quantification of transcription levels of divisome genes (**A**), MUR family genes (**B**), peptidoglycan hydrolase genes (**C**), and serine/threonine protein kinase genes (**D**). Statistical analysis was performed using *t*-test. * *p* < 0.05; ** *p* < 0.01.

**Figure 6 microorganisms-13-02245-f006:**
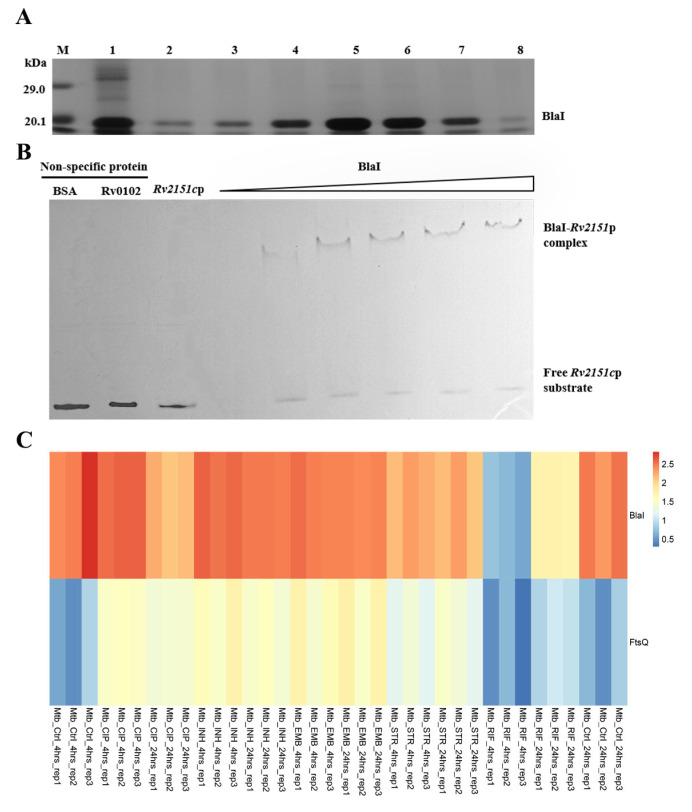
BlaI negatively regulates transcription of the cell division gene *ftsQ* (Rv2151c). (**A**) Purification of His-tagged BlaI from *E. coli* BL21. Lane 1: Protein molecular weight marker; Lanes 2–8: Eluted recombinant BlaI fractions from Ni-NTA affinity chromatography (gradient elution with increasing imidazole). The consistent ~21.1 kDa band confirms BlaI purity and stability across fractions. (**B**) EMSA validation of BlaI binding to the ftsQ (Rv2151c) promoter. Lane 1: ftsQ promoter probe only (200 ng); Lane 2: Probe + 50 ng BlaI; Lane 3: Probe + 100 ng BlaI; Lane 4: Probe + 200 ng BlaI; Lane 5: Probe + 200 ng BlaI + 10-fold excess unlabeled ftsQ promoter probe (specific competitor). Reduced probe mobility with increasing BlaI and reversal with excess unlabeled probe confirm specific binding. (**C**) Heatmap of *Rv1846c* and Rv2151c expression in GEO data of antibiotic-treated *M. tuberculosis*. Ctrl, untreated; Rep1–3, biological replicates; CIP, ciprofloxacin; INH, isoniazid; EMB, ethambutol; RIF, rifampin; STR, streptomycin.

**Figure 7 microorganisms-13-02245-f007:**
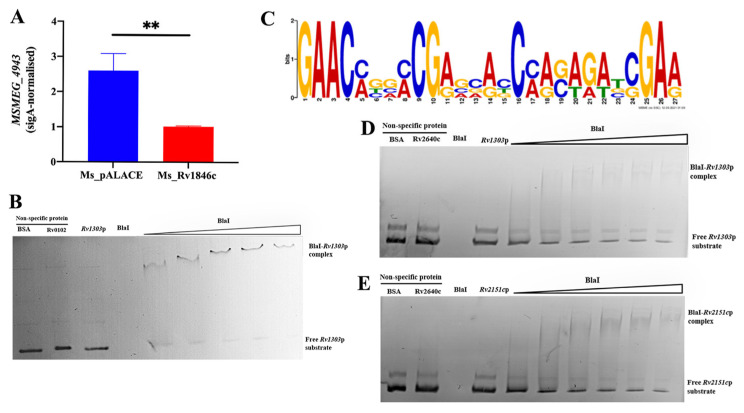
BlaI negatively regulates Rv1303, a gene co-transcribed with ATP synthase-encoding operons. (**A**) qRT-PCR analysis of *MSMEG_3630* (the *M. smegmatis* homolog of Rv1303) transcription in Ms_*Rv1846c* and Ms_pALACE. ** *p* < 0.01. (**B**) EMSA validation of BlaI binding to the Rv1303 promoter. (**C**) 30 bp promoter motifs of Rv2151c and Rv1303. Different colors correspond to specific nucleotides: red for adenine (A), green for thymine (T), blue for cytosine (C), and yellow for guanine (G). The x-axis denotes the position of nucleotides, and the y-axis represents signal intensity (bits), which reflects the confidence level of base identification. (**D**) EMSA validation of BlaI binding to the 100 bp Rv1303 promoter probe. (**E**) EMSA validation of BlaI binding to the 100 bp Rv2151c promoter probe.

**Figure 8 microorganisms-13-02245-f008:**
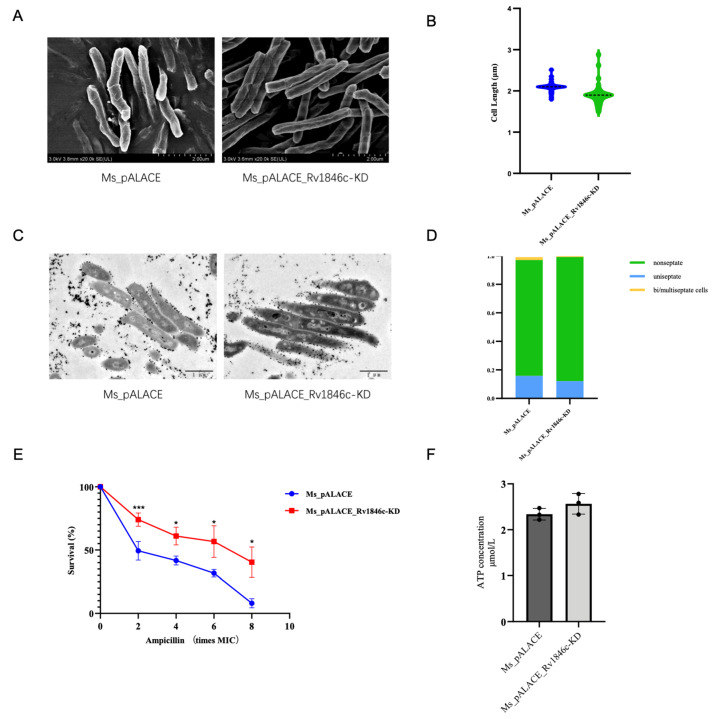
Phenotypic characterization of CRISPRi-mediated *Rv1846c* (BlaI ortholog) knockdown in *Mycobacterium smegmatis*. (**A**) Scanning Electron Microscopy (SEM) images (scale bar = 2 μm) show no obvious external morphological abnormalities in the *Rv1846c*-knockdown strain (Ms_pALACE_*Rv1846c*-KD) compared to the empty plasmid control (Ms_pALACE), unlike the elongated cells of BlaI-overexpressing strains (Figure 2A). (**B**) Cell length quantification (≥30 cells/strain, 3 biological replicates; mean ± SD, unpaired two-tailed *t*-test, *p* > 0.05) confirms no elongation in the knockdown strain (contrasting Figure 2B). (**C**) Transmission Electron Microscopy (TEM) images (scale bar = 1 μm, red arrows indicate septa) reveal similar septal formation in both strains, without the increased multi-septate cells seen in BlaI-overexpressing strains (Figure 2C). (**D**) Septum frequency analysis (300 cells/strain, 3 biological replicates; mean ± SD, unpaired two-tailed *t*-test, *p* > 0.05) shows ~18% multi-septate cells in Ms_pALACE_*Rv1846c*-KD (vs. ~17% in Ms_pALACE), inverse to the ~23% in overexpressing strains (Figure 2D). (**E**) Ampicillin killing curves (3 biological replicates; mean ± SD, two-way ANOVA, * *p* < 0.05, *** *p* < 0.001) demonstrate enhanced β-lactam resistance in the knockdown strain (~72% survival at 8× MIC vs. ~15% in Ms_pALACE), opposite to the increased sensitivity of overexpressing strains (Figure 4B). (**F**) Intracellular ATP quantification (3 biological replicates; mean ± SD, unpaired two-tailed *t*-test, *p* > 0.05) shows no significant difference between strains, in contrast to the ~42% ATP reduction in overexpressing strains (Appendix A), supporting BlaI’s specific role in ATP metabolism.

**Figure 9 microorganisms-13-02245-f009:**
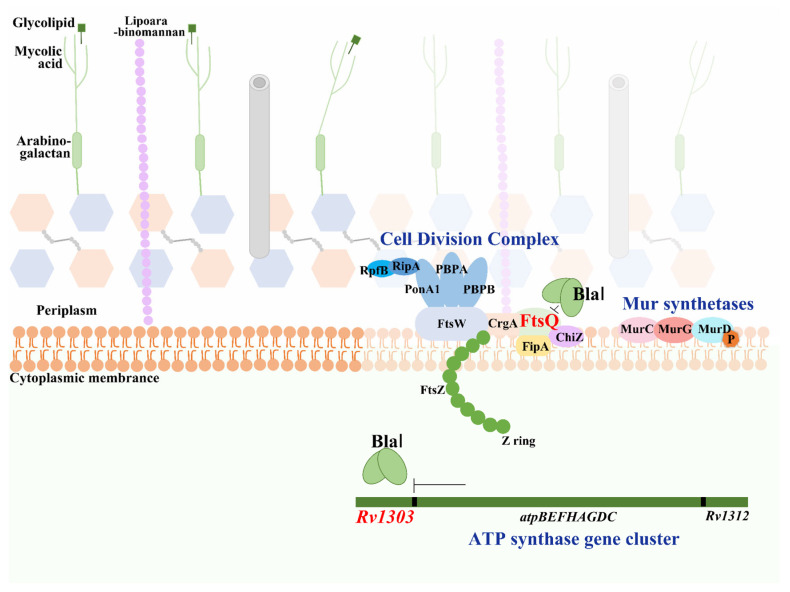
Schematic model of BlaI-mediated negative regulation of mycobacterial cell division. BlaI negatively regulates Rv1303 (co-transcribed with ATP synthase genes), reducing intracellular ATP levels. This impairs phosphorylation of bacterial division complex proteins, ultimately inhibiting cell division.

**Table 1 microorganisms-13-02245-t001:** Genes involved in *M. smegmatis* cell division.

	Gene	Product
Divisome	*ftsZ* (*MSMEG_4222*)	Cell division protein FtsZ
	*ftsQ* (*MSMEG_4225*)	Cell division protein FtsQ
	*ftsB* (*MSMEG_5414*)	septum formation initiator subfamily protein
	*ftsL* (*MSMEG_4234*)	conserved hypothetical protein
	*sepF* (*MSMEG_4219*)	conserved hypothetical protein
	*ftsI* (*MSMEG_4233*)	penicillin-binding membrane protein PbpB
	*ftsK* (*MSMEG_2690*)	DNA translocase FtsK
	*wag31* (*MSMEG_4217*)	DivIVA protein
	*ftsE* (*MSMEG_2089*)	cell division ATP-binding protein FtsE
	*ftsH* (*MSMEG_6105*)	cell division protein, ATPase
MUR family	*murC* (*MSMEG_4226*)	UDP-N-acetylmuramate--alanine ligase
	*murD* (*MSMEG_4229*)	UDP-N-acetylmuramoylalanine--D-glutamate ligase
	*murE* (*MSMEG_4232*)	UDP-N-acetylmuramoylalanyl-D-glutamate-2, 6-diaminopimelate ligas
	*murF* (*MSMEG_4231*)	UDP-N-acetylmuramoyl-tripeptide--D-alanyl-D- alanine ligase
	*murG* (*MSMEG_4227*)	undecaprenyldiphospho-muramoylpentapeptide beta-N-acetylglucosaminyltransferase
peptidoglycan hydrolase	*ripA* (*MSMEG_3145*)	Peptidoglycan hydrolase
	*cwlM* (*MSMEG_6935*)	Peptidoglycan hydrolase
	*chiZ* (*MSMEG_2742*)	Peptidoglycan hydrolase
serine/threonine protein kinase	*pknA* (*MSMEG_0030*)	serine/threonine protein kinase PknA
	*pknB* (*MSMEG_5437*)	probable serine/threonine-protein kinase PknB

## Data Availability

The original contributions presented in this study are included in the article/Appendix A. Further inquiries can be directed to the corresponding authors.

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
