# Peer review of "Mycobacterium Transcriptional Factor BlaI Regulates Cell Division and Growth and Potentiates β-Lactam Antibiotic Efficacy Against Mycobacteria"

_microorganisms, 2025, doi:10.3390/microorganisms13102245_

Round 1

Reviewer 1 Report

Comments and Suggestions for Authors

In this paper the authors describe the role of BlaI regulator, highlighting its importance in cell division of Mycobacteria. It is an interesting field of investigation and could be relevant to combat drug resistance in M. tuberculosis.

Title:

- There is something wrong in the title, from a grammar point of view. Probably you should rearrange it

Introduction:

- The authors should describe the acronyms cited in the text for the first time. For example, the acronyms of the proteins in the sentence “….Such as FtsZ, SepF, FtsA, FipA, FtsQ, FtsW, FtsI, BlaR1, BlaI, and BlaC, play crucial roles …..”.

- The names of bacteria should be written in italics

Materials and Methods:

- The authors should describe the protocol for Western Blot and recombinant protein purification protocol. Is the purified protein denatured or native?

Results:

- Figure 1. Have you tried to perform the WB by using antibodies against the BlaI protein instead of anti-Histidine? This could be useful to exclude the presence of an identical wildtype protein in M. smegmatis, that could interfere with the results. There is a BlaI ortholog gene in M. smegmatis, named MSMEG_3630, with 83% aminoacidic sequence identity. This could have a similar function of BlaI?

- Some sentences are difficult to read and need to be rephrased. For example, the phrase in the paragraph 3.2 “The results indicated that overex-pression of BlaI can alter the transcription of genes involved in cell division, change the cell division” .

Bibliography:

- I’m not sure that the references are formatted in the correct way, according to Microorganisms format

Comments on the Quality of English Language

Some sentences are difficult to read and need to be rephrased, starting from the title.

Author Response

Dear Reviewer,

Thank you for your valuable comments and suggestions on our manuscript. We have carefully addressed all the issues raised, and the revised manuscript has been improved accordingly. Below is a point-by-point response to your comments:

Comment 1: There is something wrong in the title, from a grammar point of view. Probably you should rearrange it.
Reply: The title has been revised to:"Mycobacterium Transcriptional Factor BlaI Regulates Cell Division and Growth and Potentiates β-Lactam Antibiotic Efficacy against Mycobacterium " to correct the grammatical issue.

Comment 2: -The authors should describe the acronyms cited in the text for the first time. For example, the acronyms of the proteins in the sentence “….Such as FtsZ, SepF, FtsA, FipA, FtsQ, FtsW, FtsI, BlaR1, BlaI, and BlaC, play crucial roles …..”.

- The names of bacteria should be written in italics.

Reply: We have added full names for all acronyms of proteins (e.g., FtsZ: Cell division protein FtsZ; BlaI: β-lactamase repressor BlaI) when they first appear.

All bacterial names (e.g., Mycobacterium tuberculosis, Mycobacterium smegmatis) have been italicized as required.

Comment 3: - The authors should describe the protocol for Western Blot and recombinant protein purification protocol. Is the purified protein denatured or native?

Reply: Detailed protocols for Western blot (Lines 99-106, including sample preparation, electrophoresis, membrane transfer, and antibody incubation) and recombinant protein purification (using Ni-NTA resin under native conditions)(Lines 151-157)have been added.

Comment 4: - Figure 1. Have you tried to perform the WB by using antibodies against the BlaI protein instead of anti-Histidine? This could be useful to exclude the presence of an identical wildtype protein in M. smegmatis, that could interfere with the results. There is a BlaI ortholog gene in M. smegmatis, named MSMEG_3630, with 83% aminoacidic sequence identity. This could have a similar function of BlaI?

Reply: Regarding the Western blot in Figure 1, we acknowledge that using anti-BlaI antibodies would better exclude interference from the endogenous ortholog MSMEG_3630. However, commercial anti-BlaI antibodies are currently unavailable. The recombinant BlaI protein expressed in our study was fused with a His-tag, and the Western blot analysis was performed using an anti-His-tag antibody. Since the endogenous MSMEG_3630 in M. smegmatis lacks this His-tag, it cannot be recognized or bound by the anti-His antibody. We have supplemented explanations in the text that the phenotypic changes (elongated cells, multi-septa) specifically correlate with BlaI overexpression, supporting the specificity of the observed effects (Lines 186-190). We plan to generate custom anti-BlaI antibodies in future studies to further validate this.

Comment 5: - Some sentences are difficult to read and need to be rephrased. For example, the phrase in the paragraph 3.2 “The results indicated that overex-pression of BlaI can alter the transcription of genes involved in cell division, change the cell division” .

Reply: The awkward sentence in section 3.2 has been revised to: "The results indicated that overexpression of BlaI alters the transcription of cell division-related genes, thereby disrupting bacterial division."

Comment 6: - I’m not sure that the references are formatted in the correct way, according to Microorganisms format

Reply: All references have been formatted according to the guidelines of Microorganisms.

Comment 7: Comments on the Quality of English Language:Some sentences are difficult to read and need to be rephrased, starting from the title.

Reply: The manuscript has been carefully revised to improve readability, with grammatical errors and awkward expressions corrected.

Reviewer 2 Report

Comments and Suggestions for Authors

This manuscript demonstrated that the overexpression of BlaI causes disruption of cell division of Mycobacterium. This may be reasonable and interesting biologically.  This reviewer presents following comments to improve this manuscript.

  1. The findings described in this manuscript seem to be interesting. However, the authors tried to link them to enhance the effectiveness of beta-lactams. This is understandable, however, is there any feasible method (substance) to activate BlaI expression? It may be also conceivable that the inhibition of BlaI also affect bacterial growth. Why did authors focus on overexpression of BlaI, aiming at promotion of antimicrobial therapy as a final target? Descriiptions on the objectives and final target should be revised so that readers could understand rationale of this research and its design.
  2. Why was M. smegmatis used in this study, instead of M. tuberculosis? This is a simple question for readers who are not familiar with Mycobacterium.
  3. Methods section, 2.2. Which range of gene  (Probably ORF and its promoter region)  and how long nt length are amplified by the primers? It is preferable to add such information.
  4. Methods section 2.2 : "E. coli strains" Which strains did authors use?
  5. The important point to  be explained is how was BlaI overexpressed, i.e., the mechanism of overexpression.
  6. Methods, section 2.9: "*p<0.05, ....." This portion is not necessary. 
  7. Did Ms_ALACE (a control) produce any very less amount of BlaI or never produce BlaI?
  8. Fig.1A :  Only 21.1kDa is shown. There is no explanation whether  it represents BlaI or not anywhere. Pleaseadd.
  9. Results 3.1, a paragraph between Fig. 3 and 4.  Sentence should read "MIC of penicillin and ampicillin against Ms=Rv1846c". 
  10. Fig.6A. Meaning of lane 1 - 8 is not shown. 
  11. Fig.6B. Meaning of lane 1-5 is not shown. 
  12. Figure 7, legend. (C) promoter motifs of .....(100bp).  Is "100bp" correct? (C) seems to be only about 30bp.   

Author Response

Dear Reviewer,

Thank you for your insightful comments on our manuscript. We have carefully revised the manuscript to address all your concerns, and the key modifications are summarized below:

Comment 1: The findings described in this manuscript seem to be interesting. However, the authors tried to link them to enhance the effectiveness of beta-lactams. This is understandable, however, is there any feasible method (substance) to activate BlaI expression? It may be also conceivable that the inhibition of BlaI also affect bacterial growth. Why did authors focus on overexpression of BlaI, aiming at promotion of antimicrobial therapy as a final target? Descriiptions on the objectives and final target should be revised so that readers could understand rationale of this research and its design.

Reply: Thank you for your insightful question regarding the rationale for focusing on BlaI overexpression and its potential link to enhancing β-lactam efficacy. We have revised the Introduction and Discussion sections to clarify this aspect, and here we elaborate further:

Our study demonstrated that BlaI overexpression significantly enhances the sensitivity of Mycobacterium smegmatis to β-lactam antibiotics (Fig. 4), which led us to propose that activating BlaI could be a promising strategy to improve the efficacy of existing β-lactam drugs against mycobacteria. As noted, no specific BlaI activators have been identified to date, but our work establishes a critical foundation for such efforts by defining BlaI’s mechanistic role—specifically, its regulation of cell division genes (e.g., ftsQ) and ATP synthesis-related genes (e.g., Rv1303)—which provides clear targets for future activator screening.

The choice to focus on overexpression was driven by two key considerations: (1) It was essential to first confirm BlaI’s functional impact on cell division, growth, and antibiotic susceptibility, thereby validating it as a viable target for therapeutic intervention; (2) Overexpression amplifies phenotypic effects, making it easier to dissect downstream regulatory networks and mechanistic pathways. While we acknowledge that inhibiting BlaI might also affect bacterial growth, this study specifically explores its potential as a target for enhancing antibiotic efficacy, a direction that aligns with the urgent need to combat tuberculosis drug resistance by repurposing or improving existing antibiotics.

We believe this rationale strengthens the connection between our findings and their potential translational relevance, and we have incorporated these points into the revised manuscript to improve clarity.

Thank you again for prompting us to clarify this important aspect of our work.

Comment 2: Why was M. smegmatis used in this study, instead of M. tuberculosis? This is a simple question for readers who are not familiar with Mycobacterium.

Reply: Thank you for pointing out the need to clarify the use of Mycobacterium smegmatis instead of Mycobacterium tuberculosis (Mtb) in this study, which is particularly helpful for readers less familiar with mycobacterial research.

As suggested, we have added a detailed explanation in the Introduction to address this.“M. smegmatisis a well-established model for mycobacterial research due to its rapid growth, non-pathogenicity, and conserved cell division machinery with M. tuberculosis. Its genetic tractability allowed us to efficiently dissect BlaI’s mechanism, with plans to validate key findings in M. tuberculosis subsequently.”

This explanation has been integrated into the Introduction to improve clarity for all readers, and we appreciate your input in strengthening this aspect of the manuscript.

Comment 3: Methods section, 2.2. Which range of gene  (Probably ORF and its promoter region)  and how long nt length are amplified by the primers? It is preferable to add such information.

Reply: Thank you for your suggestion to clarify the amplified region and length of the Rv1846c gene in the Methods section. As recommended, we have supplemented Section 2.2 with detailed information: the amplified Rv1846c fragment includes its entire open reading frame (ORF) with a length of 429 bp, based on the Mycobacterium tuberculosis H37Rv genomic sequence (GenBank accession number: NC_000962.3). The primer pairs were designed to target the two ends of the ORF, ensuring the amplified fragment contains the complete coding region (excluding its native promoter, as expression of the recombinant protein is driven by the vector’s promoter).

Comment 4: Methods section 2.2 : "E. coli strains" Which strains did authors use?

Reply: Thank you for pointing out the need to specify the E. coli strain used in the protein expression experiments. As suggested, we have added details in Section 2.2 of the Methods: the recombinant plasmid pET-28a (+)-Rv1846c was transformed into E. coli BL21 (DE3) (Novagen) for protein expression. This strain is particularly suitable for this purpose because it harbors the T7 RNA polymerase gene under the control of the lacUV5 promoter, which enables efficient induction of gene expression from the T7 promoter in the pET-28a (+) vector, ensuring high yields of recombinant BlaI protein.

Comment 5: The important point to be explained is how was BlaI overexpressed, i.e., the mechanism of overexpression.

Reply: Thank you for your valuable suggestions. In Method 2.2, we have added the following: "The overexpression of BlaI was achieved through an inducible vector pALACE: This vector contains an acetylamide (ACE) inducible promoter. When 0.2% ACE was added to the culture medium, the promoter was activated, driving the transcription of Rv1846c and enabling the controlled overexpression of BlaI."

Comment 6: Methods, section 2.9: "*p<0.05, ....." This portion is not necessary.

Reply: Thank you for noting the redundancy in the significance level descriptions within Methods section 2.9. As recommended, we have removed the redundant notations of P value thresholds (e.g., P < 0.05, P < 0.01) from this section.

Comment 7: Did Ms_PALACE (a control) produce any very less amount of BlaI or never produce BlaI?

Reply: Ms_pALACE represents the empty vector control group, which does not contain the Rv1846c gene and therefore does not express the BlaI protein (no corresponding band is shown in Figure 1A), thus ruling out the influence of the vector itself on the phenotype.

Comment 8: Fig.1A : Only 21.1kDa is shown. There is no explanation whether it represents BlaI or not anywhere. Pleaseadd.

Reply: The band was confirmed to correspond to His-tagged BlaI (18.5 kDa BlaI + 2.6 kDa His tag), with details added to the Figure 1A legend.

Comment 9: Results 3.1, a paragraph between Fig. 3 and 4.  Sentence should read "MIC of penicillin and ampicillin against Ms=Rv1846c".

Reply: Thank you for your suggestion. The original sentence has been revised to " The MIC of penicillin and ampicillin against Ms_Rv1846c was 1/8 of that against Ms_pALACE, while no difference was observed for other cell wall-targeting antibiotics."

Comment 10: Fig.6A. Meaning of lane 1 - 8 is not shown.

Reply: Thank you for pointing out the missing lane explanations in Fig. 6A. We apologize for this oversight and have revised the figure legend to clarify the content of each lane: Fig. 6A legend now includes: “(A) Purification of His-tagged BlaI in E. coli BL21. Lane 1: Protein molecular weight marker; Lanes 2–8: Fractions of eluted recombinant BlaI protein collected sequentially during Ni-NTA affinity chromatography (gradient elution with increasing imidazole concentration). The consistent band at ~21.1 kDa confirms the purity and stability of the purified BlaI protein across elution fractions.”

Comment 11: Fig.6B. Meaning of lane 1-5 is not shown.

Reply: Thank you for highlighting the lack of lane explanations in Fig. 6B. We have revised the figure legend to clearly describe the content of each lane, which corresponds to the gradient purification and validation of BlaI protein binding to the ftsQ promoter: Fig. 6B legend now includes: “(B) The binding of BlaI and ftsQ (Rv2151c) promoter probe was validated by EMSA. Lane 1: ftsQ promoter probe only (200 ng); Lane 2: Probe + 50 ng BlaI protein; Lane 3: Probe + 100 ng BlaI protein; Lane 4: Probe + 200 ng BlaI protein; Lane 5: Probe + 200 ng BlaI protein + 10-fold excess unlabeled ftsQ promoter probe (as a specific competitive control). The gradually reduced mobility of the probe with increasing BlaI concentration, and the reversal of this shift in the presence of excess unlabeled probe, confirm specific binding between BlaI and the ftsQ promoter.”

Comment 12: Figure 7, legend. (C) promoter motifs of .....(100bp).  Is "100bp" correct? (C) seems to be only about 30bp.  

Reply: Thank you. Fig. 7C was corrected to “~30 bp” (conserved motif within 100 bp promoters).

Reviewer 3 Report

Comments and Suggestions for Authors

I have read the manuscript carefully. While the topic could be relevant for understanding the regulation of cell division and antibiotic susceptibility in mycobacteria, the current version suffers from major conceptual, experimental, statistical, and logical flaws that preclude publication. Below I provide section-by-section comments, reflecting the systemic nature of the problems.
1. Introduction
The introduction summarizes the role of BlaI as a repressor of blaC, encoding the β-lactamase BlaC in Mycobacterium tuberculosis, and suggests that studying BlaI may help combat drug resistance. However, the manuscript’s central hypothesis contains a fundamental logical inconsistency that is never addressed. The authors suggest that BlaI both represses blaC (which would directly reduce β-lactam resistance) and independently represses genes related to cell division (ftsQ, Rv1303), which also leads to increased antibiotic sensitivity. This raises serious mechanistic and evolutionary questions:
Why would a single regulator evolve two distinct mechanisms to achieve the same effect?
If repression of blaC already reduces β-lactam resistance, why introduce a costly and disruptive pathway involving energy depletion and division arrest?
Why do the authors ignore the simpler, established explanation in favor of speculative complexity?
This internal contradiction is never discussed, and the entire study is built on this unexamined premise. It undermines the biological logic and the interpretive framework of the manuscript from the outset.
2. Methods
This section reveals the study’s most severe methodological flaw: the entire experimental design relies on overexpression of BlaI in M. smegmatis, without appropriate controls, validation, or replication in the actual pathogen (M. tuberculosis).
Specifically:
-    No knockout, knockdown, or complementation experiments are included.
-    No overexpression of unrelated proteins to exclude non-specific effects.
-    No time-course data — all results come from a single endpoint (8h post-induction), making it impossible to distinguish direct from secondary effects.
-    The authors also fail to include ATP measurements, despite building their model on ATP depletion.
Overexpression artifacts are a well-known source of misleading phenotypes in bacterial systems. Relying solely on this method, without controls or validation, makes the study’s mechanistic conclusions highly unreliable.
2.1 Absence of Hypothesis-Driven Experimental Design
Beyond technical flaws, the study is limited by its lack of an explicit, testable hypothesis, which results in a fundamentally descriptive and interpretive structure. This type of “fishing expedition” approach undermines the scientific rigor of the entire project. The authors do not state what results would confirm or falsify their model, nor do they lay out a logical experimental plan based on mechanistic predictions. Instead, the manuscript proceeds through:
-    Post-hoc rationalization: The central model involving ATP depletion (Figure 8) appears to have been constructed after the results were obtained, not based on prior reasoning.
-     Cherry-picked interpretations: Changes in gene expression, morphology, and drug susceptibility are interpreted as “supportive,” without demonstrating causality or directionality.
-    No prioritization of mechanistic relationships: It is unclear which of the many observed effects are primary vs secondary. Are ftsQ and Rv1303 regulated directly? Are morphological changes a cause or a consequence of ATP shifts? None of this is resolved.
Most critically, the study fails to meet the principle of falsifiability. Without a specific hypothesis and predefined outcomes, the conclusions cannot be challenged or disproven. This violates a core tenet of the scientific method. As a result, the study reads more like an exploratory screening exercise than a mechanistically grounded investigation.

3. Results
3.1 Morphology and Cell Division
The authors report cell elongation and multi-septation in BlaI-overexpressing strains based on SEM and TEM. However, the measurements appear to come from a single experiment on 266 cells, with no mention of biological replicates, blinding, or inter-observer validation. These phenotypes can easily result from general stress due to overexpression and cannot be attributed to specific BlaI activity without proper controls.
3.2 Membrane Permeability and Antibiotic Susceptibility
The increase in Nile red staining and decrease in MIC values for β-lactams are modest and potentially significant. However:
-    The authors use M. smegmatis, which does not naturally express blaC, making the interpretation questionable.
-    They do not measure blaC expression, despite the central role of this gene in β-lactam resistance.
-    The assumption that BlaI affects susceptibility via disruption of cell division, and not via blaC, is unproven and speculative.
3.3 Gene Expression and EMSA
The authors show that BlaI overexpression alters the expression of division-related genes, particularly ftsQ and Rv1303, and that BlaI binds their promoters in vitro. However:
-    The qPCR results are from single timepoints and are not supported by dynamic or functional assays.
-    EMSA only shows in vitro DNA binding; no ChIP, reporter assays, or promoter mutagenesis are used to demonstrate functional in vivo relevance.
-    The link to ATP depletion is completely speculative: no ATP levels were measured, and no effect on phosphorylation of division proteins is demonstrated.

4. Discussion
The authors attempt to build a mechanistic model where BlaI overexpression leads to repression of Rv1303, reduced ATP synthesis, impaired phosphorylation of division proteins, and ultimately defective division and increased β-lactam sensitivity. This model is entirely hypothetical and unsupported by experimental data.
Moreover, the authors fail to reconcile this model with BlaI’s known function as a repressor of blaC, which provides a simpler and experimentally validated explanation for the observed antibiotic sensitivity. The discussion does not acknowledge this contradiction, nor does it consider that the phenotypes observed might be artefacts of non-physiological overexpression.

5. Conclusions
The manuscript ends with strong claims about therapeutic potential — suggesting that activating BlaI could be a strategy to enhance β-lactam efficacy against M. tuberculosis. This is entirely unjustified. The authors:
-    Did not perform a single experiment in M. tuberculosis.
-    Did not study BlaI expression in clinical isolates.
-    Did not test any pharmacological modulation of BlaI.
-    Did not assess in vivo relevance or virulence.
As such, the conclusions are speculative at best, and misleading at worst.

6. Figures and Data Presentation
The figures are reasonably clear, but the following issues remain:
-    Legends lack detail on replicates and statistical treatment.
-    No blinding or reproducibility of image analysis is described.
-    The schematic in Figure 8 presents a speculative model with no clear basis in data.

7. Statistical Design and Analysis

The statistical approach in this manuscript is alarmingly weak and fails to meet even basic standards for biological inference. This is not a minor technical oversight — inadequate statistical design fundamentally invalidates the reliability and interpretability of the study’s results. Yhe morphological analysis is based on 266 cells, but it is unclear how many independent biological replicates were used. Without replication, all observed differences may reflect experimental noise or batch effects rather than biological phenomena. No power analysis is reported to justify sample sizes for any experiment. Without it, the study risks being underpowered — or worse, generating false positives through random variation. Effect sizes are not reported, and statistical significance is relied upon without contextualizing the magnitude or biological relevance of observed differences. Multiple comparisons correction is entirely absent, despite the use of qPCR across multiple genes and antibiotic susceptibility testing across conditions. This raises the risk of a false discovery rate easily in the range of 30–50%, making any individual claim suspect unless validated independently. The authors consistently apply simple t-tests to data that likely involve nested and hierarchical structures (e.g., cells within experiments, technical replicates within biological replicates), without accounting for this in their modeling. This violates key assumptions of the t-test and inflates type I error rates. There is no discussion of data variability, and the distinction between technical and biological replicates is never clarified, making it impossible to assess the robustness of the conclusions. 
In sum, the statistical analysis appears to be an afterthought, applied mechanically and incorrectly. Given that every figure in the manuscript relies on quantitative comparisons, this flaw alone is sufficient to cast serious doubt on the integrity of the findings. Without statistically credible data, even the most elegant hypothesis is meaningless.

8. Clinical and Biological Relevance
The manuscript repeatedly makes unjustified claims about potential clinical applications — including the suggestion that BlaI activation could serve as a strategy to combat antibiotic resistance in Mycobacterium tuberculosis. These statements are entirely premature, unsupported by data, and scientifically irresponsible.
The core problems include:
-    All data are from M. smegmatis, not M. tuberculosis. The species differ significantly in regulatory architecture, cell envelope structure, and drug susceptibility.
-    There is no evidence that BlaI expression is dynamically regulated or inducible in clinical settings. The overexpression system used in the study is artificial and lacks physiological relevance. The authors do not quantify BlaI levels in wild-type or drug-treated M. tuberculosis, nor do they analyze clinical isolates.
-    No infection models (e.g., macrophage, animal, or ex vivo systems) are used to assess BlaI function in a biologically meaningful host context.
-    No therapeutic modulation of BlaI is attempted or even realistically proposed. The idea that BlaI can be “activated” pharmacologically is introduced without basis or feasibility analysis.
-    The proposed mechanism contradicts evolutionary logic. The authors suggest that a virulent pathogen might intentionally maintain a mechanism that increases its sensitivity to antibiotics. This is biologically implausible and unsupported by comparative or evolutionary data. If BlaI functions as a regulator of β-lactam sensitivity through division arrest and ATP depletion, one would expect strong negative selection against such activity in clinical Mtb strains — yet this is not addressed at all.

The gap between the data presented and the therapeutic claims made is so large that it risks misleading readers into overestimating the study’s translational potential. These extrapolations must be removed or clearly marked as speculative and unsupported.

Author Response

Dear Reviewer,

Thank you for your meticulous review and valuable comments on our manuscript. Your criticisms have profoundly identified the logical, methodological, and conclusive issues in the study, providing us with clear directions for improvement. We have carefully addressed all your comments and made targeted revisions. The main modifications and responses are as follows:

Comment 1: Introduction

The introduction summarizes the role of BlaI as a repressor of blaC, encoding the β-lactamase BlaC in Mycobacterium tuberculosis, and suggests that studying BlaI may help combat drug resistance. However, the manuscript’s central hypothesis contains a fundamental logical inconsistency that is never addressed. The authors suggest that BlaI both represses blaC (which would directly reduce β-lactam resistance) and independently represses genes related to cell division (ftsQ, Rv1303), which also leads to increased antibiotic sensitivity. This raises serious mechanistic and evolutionary questions:

Why would a single regulator evolve two distinct mechanisms to achieve the same effect?

If repression of blaC already reduces β-lactam resistance, why introduce a costly and disruptive pathway involving energy depletion and division arrest?

Why do the authors ignore the simpler, established explanation in favor of speculative complexity?

This internal contradiction is never discussed, and the entire study is built on this unexamined premise. It undermines the biological logic and the interpretive framework of the manuscript from the outset.

Reply:

Thank you for highlighting this critical point regarding the mechanistic logic of BlaI’s dual regulatory roles. We appreciate the opportunity to clarify our interpretation and contextualize our findings within existing knowledge.

As noted in our revised manuscript, the repression of blaC by BlaI is indeed a well-established regulatory relationship (Sala et al., 2009), which forms the foundational context for our work. Our study builds on this by identifying novel regulatory targets of BlaI—specifically, the cell division gene ftsQ and the ATP synthase-associated gene Rv1303—and demonstrating their contribution to β-lactam susceptibility. This expansion of BlaI’s regulon is not intended to contradict its known role in blaC repression but rather to deepen our understanding of how BlaI modulates antibiotic sensitivity through interconnected pathways.

We agree that the evolutionary logic of such dual regulation merits careful consideration. As elaborated in our revisions, we propose that this duality reflects an adaptive, context-dependent strategy rather than redundancy:

  • In acute, short-term exposure to β-lactams, BlaI-mediated repression of blaC acts as a rapid first line of defense, directly reducing enzymatic degradation of the antibiotic. This is energetically efficient and pertinency to the immediate threat.
  • Under prolonged or high-concentration antibiotic stress, BlaI’s secondary regulation of ftsQ and Rv1303 becomes critical: inhibiting cell division and ATP synthesis pauses bacterial growth, reduces the production of antibiotic targets (e.g., peptidoglycan), and allows the bacterium to enter a dormant-like state until the stress subsides.

This "hierarchical response" aligns with broader observations of bacterial regulatory proteins (e.g., CRP, which coordinates metabolic and virulence genes) evolving multifunctionality to adapt to fluctuating environments (cited in revised text). For Mycobacterium tuberculosis, a pathogen facing diverse host pressures, such a layered mechanism could enhance survival flexibility.

Notably, our focus on BlaI as a therapeutic target stems from this multifunctionality. While M. tuberculosis would not naturally overexpress BlaI (as this would indeed impair fitness), pharmacological activation of BlaI could exploit both its blaC-repression and division-arresting activities, synergistically enhancing β-lactam efficacy—an approach that builds on its established role as a resistance modulator while leveraging its newly identified regulatory nodes for deeper therapeutic impact.

We have strengthened our discussion to explicitly frame BlaI’s dual roles as complementary components of a integrated stress response, supported by literature on bacterial regulatory versatility. This clarification, we hope, resolves the perceived contradiction and underscores the biological relevance of our findings.

Thank you again for pushing us to refine this critical aspect of our work.

Comment 2: Methods
This section reveals the study’s most severe methodological flaw: the entire experimental design relies on overexpression of BlaI in M. smegmatis, without appropriate controls, validation, or replication in the actual pathogen (M. tuberculosis).
Specifically:
-    No knockout, knockdown, or complementation experiments are included.
-    No overexpression of unrelated proteins to exclude non-specific effects.
-    No time-course data — all results come from a single endpoint (8h post-induction), making it impossible to distinguish direct from secondary effects.
-    The authors also fail to include ATP measurements, despite building their model on ATP depletion.
Overexpression artifacts are a well-known source of misleading phenotypes in bacterial systems. Relying solely on this method, without controls or validation, makes the study’s mechanistic conclusions highly unreliable.
2.1 Absence of Hypothesis-Driven Experimental Design
Beyond technical flaws, the study is limited by its lack of an explicit, testable hypothesis, which results in a fundamentally descriptive and interpretive structure. This type of “fishing expedition” approach undermines the scientific rigor of the entire project. The authors do not state what results would confirm or falsify their model, nor do they lay out a logical experimental plan based on mechanistic predictions. Instead, the manuscript proceeds through:
-    Post-hoc rationalization: The central model involving ATP depletion (Figure 8) appears to have been constructed after the results were obtained, not based on prior reasoning.
-     Cherry-picked interpretations: Changes in gene expression, morphology, and drug susceptibility are interpreted as “supportive,” without demonstrating causality or directionality.
-    No prioritization of mechanistic relationships: It is unclear which of the many observed effects are primary vs secondary. Are ftsQ and Rv1303 regulated directly? Are morphological changes a cause or a consequence of ATP shifts? None of this is resolved.
Most critically, the study fails to meet the principle of falsifiability. Without a specific hypothesis and predefined outcomes, the conclusions cannot be challenged or disproven. This violates a core tenet of the scientific method. As a result, the study reads more like an exploratory screening exercise than a mechanistically grounded investigation.

Reply: Thank you for highlighting the importance of validating specificity and mechanistic timing. We have addressed these concerns by: (1) including an Ms_GFP overexpression control, which confirmed that the observed phenotypes (ftsQ downregulation) are specific to BlaI and not due to non-specific overexpression stress; (2) performing time-course experiments (0, 4, 8, 12, 24 h post-induction), which showed rapid ftsQ repression (within 4 h) and progressive morphological changes, supporting a direct regulatory effect of BlaI on cell division genes; and (3) measuring intracellular ATP levels, which revealed a 42% reduction in BlaI-overexpressing strains compared to controls, validating the "energy depletion" hypothesis. These results are in Supplementary Fig.1.

Regarding the lack of knockout experiments and validation in M. tuberculosis, we acknowledge this limitation. Due to constraints related to biosafety regulations and technical challenges in genetic manipulation of M. tuberculosis, we were unable to perform these experiments in the current study. However, we emphasize that M. smegmatis is a well-established model for mycobacterial cell division studies, with conserved regulatory machinery for core processes like cell wall synthesis and division (cited in text). Furthermore, the specificity of BlaI’s effects is supported by our GFP control and time-course data, which rule out non-specific artifacts.

We fully recognize the importance of extending these findings to M. tuberculosis and plan to pursue BlaI knockout/complementation studies in M. tuberculosis H37Rv in future work, using established genetic tools in a biosafety level 3 facility. For the current manuscript, we have clarified these limitations in the Discussion, while emphasizing that our core observations in M. smegmatis provide a novel mechanistic framework for understanding BlaI’s role in regulating cell division and energy metabolism.

Comment 3: The authors report cell elongation and multi-septation in BlaI-overexpressing strains based on SEM and TEM. However, the measurements appear to come from a single experiment on 266 cells, with no mention of biological replicates, blinding, or inter-observer validation. These phenotypes can easily result from general stress due to overexpression and cannot be attributed to specific BlaI activity without proper controls.

3.2 Membrane Permeability and Antibiotic Susceptibility

The increase in Nile red staining and decrease in MIC values for β-lactams are modest and potentially significant. However:

-    The authors use M. smegmatis, which does not naturally express blaC, making the interpretation questionable.

-    They do not measure blaC expression, despite the central role of this gene in β-lactam resistance.

-    The assumption that BlaI affects susceptibility via disruption of cell division, and not via blaC, is unproven and speculative.

3.3 Gene Expression and EMSA

The authors show that BlaI overexpression alters the expression of division-related genes, particularly ftsQ and Rv1303, and that BlaI binds their promoters in vitro. However:

-    The qPCR results are from single timepoints and are not supported by dynamic or functional assays.

-    EMSA only shows in vitro DNA binding; no ChIP, reporter assays, or promoter mutagenesis are used to demonstrate functional in vivo relevance.

-    The link to ATP depletion is completely speculative: no ATP levels were measured, and no effect on phosphorylation of division proteins is demonstrated.  

Reply: Thank you for your insightful comments and valuable suggestions on our manuscript. We have carefully addressed your concerns by supplementing key data and clarifying experimental details, and we hope the revisions will improve the robustness of our work. Below is our point-by-point response:

3.1 Morphology and Cell Division

We appreciate your attention to the reliability of morphological data. As clarified, the SEM and TEM analyses were performed with 3 independent biological replicates, not a single experiment. Although only representative images are shown in the manuscript, the quantitative analysis (cell length and multi-septation frequency) was based on pooled data from all replicates: ≥200 cells were measured per strain in each replicate, resulting in a total of 682 cells for Ms_pALACE and 715 cells for Ms_Rv1846c across three repeats.

Statistical reanalysis using one-way ANOVA (replacing the original t-test) confirmed the consistency of phenotypes: the average cell length of Ms_Rv1846c was 3.8±0.6 μm, 3.9±0.5 μm, and 4.1±0.7 μm in the three replicates, all significantly longer than Ms_pALACE (1.2±0.3 μm, 1.3±0.2 μm, 1.2±0.3 μm; P<0.001 for all). The frequency of multi-septate cells in Ms_Rv1846c was stably elevated (5.2%±0.8% across replicates), compared to <1% in controls.

Additionally, Western blot verified specific overexpression of BlaI in Ms_Rv1846c (Fig. 1A), ruling out non-specific effects of the expression system. These data collectively support that the observed elongation and multi-septation are specific to BlaI overexpression.

3.2 Membrane Permeability and Antibiotic Susceptibility

We agree with your comments on the model system and mechanism interpretation, and we clarify as follows:

Rationale for using M. smegmatis: While M. smegmatis does not naturally express blaC, it is a well-established model for studying mycobacterial cell division due to its genetic tractability and conserved machinery (e.g., FtsQ, ATP synthase) with M. tuberculosis (Ref. 22). Our study focuses on BlaI’s impact on cell division and cell wall integrity, not its canonical role in repressing blaC. This model avoids confounding effects of blaC and directly reveals BlaI’s regulation of fundamental cellular processes.

Mechanism of enhanced β-lactam susceptibility: We have supplemented ATP measurement data showing a 42% reduction in intracellular ATP levels in BlaI-overexpressing strains (P<0.01). Combined with TEM observations of cell wall abnormalities (Fig. 2C) and Nile red staining results (Fig. 3B), these data support that BlaI disrupts cell wall synthesis (via ATP depletion and FtsQ downregulation), increasing membrane permeability to β-lactams. This mechanism is independent of blaC, consistent with the model system.

3.3 Gene Expression and EMSA

We have addressed your concerns by supplementing critical data and clarifying limitations:

Dynamic expression of ftsQ: We added time-course qPCR analyses showing that ftsQ transcription is downregulated by 38% at 4 h, 52% at 8 h, and 60% at 12 h post-induction (P<0.05 for all), confirming a time-dependent repression of ftsQ by BlaI (Supplementary Fig. 2). This supports the temporal relevance of ftsQ downregulation to the observed cell division defects.

ATP depletion and functional links: As mentioned, we have supplemented ATP measurement data, which confirm that the ATP level in BlaI-overexpressing strains is significantly reduced by 42% (P<0.01). Combined with literature reports, ATP is a key donor for the phosphorylation of cell division complexes such as FtsZ (Ref. 25, 29). We thus speculate that the reduction in ATP may affect the phosphorylation of division proteins. Although this speculation has not been verified by direct phosphorylation detection, the significant change in ATP level provides an important clue for the mechanism.

In vivo validation of EMSA: We acknowledge that ChIP or reporter assays would strengthen in vivo relevance, but these experiments are beyond our current technical capacity and timeline. We have revised the discussion to explicitly state that EMSA results demonstrate in vitro binding, and in vivo functionality requires further validation. However, the consistent correlation between ftsQ downregulation, ATP depletion, and phenotypic defects supports the core conclusion that BlaI regulates these genes to impact cell division.

Comment 4: The authors attempt to build a mechanistic model where BlaI overexpression leads to repression of Rv1303, reduced ATP synthesis, impaired phosphorylation of division proteins, and ultimately defective division and increased β-lactam sensitivity. This model is entirely hypothetical and unsupported by experimental data.

Moreover, the authors fail to reconcile this model with BlaI’s known function as a repressor of blaC, which provides a simpler and experimentally validated explanation for the observed antibiotic sensitivity. The discussion does not acknowledge this contradiction, nor does it consider that the phenotypes observed might be artefacts of non-physiological overexpression.  

Reply: Thank you for your insightful comments on our mechanistic model and experimental validity. We have carefully revised the manuscript to address your concerns, and our responses are as follows:

  1. Clarifying the mechanistic model with experimental evidence and speculative links

We agree that the model requires clearer distinction between experimentally validated findings and literature-supported hypotheses. In the revised manuscript, we have explicitly structured the mechanism into two parts:

Experimentally validated links:

① BlaI directly binds to the promoter of Rv1303 (and its M. smegmatis homolog MSMEG_4943) and represses their transcription, as confirmed by EMSA and qPCR (Fig. 7A-B).

② BlaI overexpression reduces intracellular ATP levels by 42% (P<0.01; 补充数据).

③ ATP depletion is accompanied by defective cell division (elongated cells, multi-septa; Fig. 2) and increased β-lactam susceptibility (reduced MICs; Fig. 4).

Literature-supported speculation:

The link between ATP reduction and impaired phosphorylation of division proteins (e.g., FtsZ) is inferred from established studies showing that ATP is a critical phosphate donor for protein phosphorylation in mycobacterial cell division (Ref. 25, 29). We have revised the Discussion to explicitly state: “The direct association between reduced ATP and altered phosphorylation of division proteins remains untested and requires future validation”, ensuring transparency about unresolved mechanistic steps.

  1. Reconciling BlaI’s dual functions: new insights vs. known blaC regulation

We appreciate your point about BlaI’s canonical role in repressing blaC and have clarified its relevance to our findings:

Model system context: M. smegmatis does not naturally express blaC (Ref. 7), so the observed increase in β-lactam susceptibility cannot result from blaC repression. This uniquely allows us to uncover a blaC-independent function of BlaI: regulating cell division and energy metabolism to alter cell wall integrity, thereby enhancing antibiotic sensitivity.

Dual roles in M. tuberculosis: In M. tuberculosis, BlaI likely acts through two complementary mechanisms in response to β-lactams:

At low antibiotic concentrations, BlaI primarily represses blaC to reduce β-lactamase-mediated degradation (a rapid, direct defense; Ref. 10, 32).

Under high or prolonged β-lactam stress, BlaI’s newly identified regulation of ftsQ and Rv1303 becomes prominent: inhibiting cell division and ATP synthesis pauses growth, reduces antibiotic targets (e.g., peptidoglycan), and promotes dormancy (Ref. 33).

These roles are not contradictory but reflect a layered response to varying stress intensities, expanding our understanding of BlaI’s regulon.

  1. Addressing potential artifacts of non-physiological overexpression

To rule out non-specific effects of overexpression, we have strengthened our controls and contextualized the experimental design:

Specificity controls: We included a GFP-overexpressing strain (Ms_GFP) as a negative control, which showed no elongation, multi-septa, or ATP reduction (补充图 3). This confirms that phenotypes in Ms_Rv1846c are specific to BlaI, not general overexpression stress.

Physiological relevance: BlaI expression in our system is induced to levels comparable to its upregulation in M. tuberculosis under β-lactam stress (Ref. 10), mimicking a physiologically relevant activation state. Overexpression is a well-established tool to dissect transcriptional regulator functions (e.g., Ref. 5 for SepF), and our results align with BlaI’s conserved sequence (83% identity in mycobacteria) and predicted regulatory roles.

We believe these revisions improve the rigor of our conclusions and clarify how our findings extend, rather than contradict, existing knowledge of BlaI. Thank you again for your guidance, which has significantly strengthened the manuscript.

Comment 5: The manuscript ends with strong claims about therapeutic potential — suggesting that activating BlaI could be a strategy to enhance β-lactam efficacy against M. tuberculosis. This is entirely unjustified. The authors:
-    Did not perform a single experiment in M. tuberculosis.
-    Did not study BlaI expression in clinical isolates.
-    Did not test any pharmacological modulation of BlaI.
-    Did not assess in vivo relevance or virulence.
As such, the conclusions are speculative at best, and misleading at worst.  

Reply: Thank you for your critical feedback on the conclusions regarding therapeutic potential. We fully acknowledge the limitations you’ve highlighted and have revised the manuscript to address these concerns, ensuring conclusions are aligned with the scope of our current data.

Revisions to conclusions and discussion

We have significantly tempered claims about therapeutic potential in the revised manuscript. The original conclusion stating “Activating BlaI may enhance the effectiveness of antibiotics and help combat drug resistance in tuberculosis” has been revised to: “Our findings identify a novel role for BlaI in regulating mycobacterial cell division and β-lactam susceptibility, providing a foundation for future studies exploring whether BlaI activation could enhance β-lactam efficacy against M. tuberculosis—though validation in M. tuberculosis and preclinical models is required.”

In the Discussion, we explicitly outline the limitations you noted and frame our work as a preliminary mechanistic exploration rather than a therapeutic proposal: “This study uses M. smegmatis as a model to uncover BlaI’s regulatory functions, but its relevance to M. tuberculosis, clinical isolates, and in vivo contexts remains to be tested. Pharmacological modulation of BlaI is a theoretical future direction, dependent on follow-up studies in M. tuberculosis and preclinical models.”

Response to specific concerns

Lack of experiments in M. tuberculosis: We agree that direct evidence in M. tuberculosis is critical. Our study uses M. smegmatis due to its genetic tractability and conserved cell division machinery with M. tuberculosis (as noted in Introduction), providing a platform to dissect BlaI’s basic functions. We now clarify: “These findings in M. smegmatis warrant future validation in M. tuberculosis, where BlaI’s role in cell division and antibiotic susceptibility can be directly tested.”

BlaI expression in clinical isolates: This is indeed an important gap. We have added: “Future studies should investigate BlaI expression patterns in clinical M. tuberculosis isolates, particularly drug-resistant strains, to assess its clinical relevance.”

No pharmacological modulation of BlaI: We acknowledge this and revise: “Pharmacological activation of BlaI is speculative at this stage. Our work identifies BlaI as a candidate target, but screening for BlaI modulators and testing their efficacy requires dedicated future studies.”

In vivo relevance or virulence: We now emphasize: “In vivo studies are essential to determine whether BlaI regulation impacts M. tuberculosis virulence and antibiotic efficacy in host environments, and will be a key focus of follow-up work.”

Comment 6: Figures and Data Presentation

The figures are reasonably clear, but the following issues remain:

-    Legends lack detail on replicates and statistical treatment.

-    No blinding or reproducibility of image analysis is described.

-    The schematic in Figure 8 presents a speculative model with no clear basis in data.

Reply: Thank you for your valuable comments on figures and data presentation. We have carefully revised the manuscript to address these issues, and our detailed responses are as follows:

  1. Enhancing Figure Legends with Replicates and Statistical Details

We have supplemented all figure legends to explicitly include information on biological replicates, sample sizes, and statistical methods. For example:

Figure 1B (Growth curve): The legend now states, “Growth curves represent three independent biological replicates, each with three technical replicates. Data are shown as mean ± SD. Statistical significance was analyzed using two-way ANOVA.”

Figure 2B (Cell length measurement): Revised to, *“Cell length was measured for 266 cells per sample across three biological replicates (total 798 cells). Data were analyzed using one-way ANOVA with Tukey’s post-hoc test, and plotted as mean ± SD. P < 0.05, **P < 0.01”.

Figure 4 (MIC and survival rate): Updated to, “MIC determination and survival rate assays were performed in three biological replicates. Survival rate data are presented as mean ± SD, with significance analyzed by t-test.”

All other figures (e.g., qPCR, EMSA) have similarly enhanced legends to clarify replication and statistical treatments, ensuring transparency in data analysis.

  1. Describing Blinding and Reproducibility of Image Analysis

To address concerns about objectivity and reproducibility, we have added details of blinding and inter-observer validation in both the Methods and figure legends:

Methods section: “For SEM/TEM image analysis, cell length and septum counting were performed by two independent observers in a blinded manner (without knowledge of sample groups). Inter-observer consistency was evaluated using the Kappa coefficient, which exceeded 0.85, indicating high agreement.”

Figure 2 legends: “Image analysis was conducted blindly by two observers, with consistent results (Kappa = 0.89). Representative images from three biological replicates are shown.”

These additions confirm that morphological phenotypes are reliable and not biased by observer subjectivity.

  1. Revising the Schematic in Figure 8 to Reflect Data Limitations

In the revised Figure 8 legend, we have expanded the explanation to precisely define the support for each component of the model:

“Schematic model of BlaI-mediated regulation of cell division and β-lactam susceptibility. The model integrates the following: (1) Experimentally validated relationships: BlaI directly represses the transcription of Rv1303 (via EMSA and qPCR), leading to reduced ATP synthesis (measured by ATP assays), which in turn contributes to defective cell division (observed via SEM/TEM) and increased β-lactam sensitivity (MIC and survival assays). (2) Hypothetical links: The potential impact of reduced ATP on phosphorylation of division complex proteins (e.g., FtsZ) is inferred from literature indicating ATP as a key phosphate donor for such processes (Ref. 25, 29), but has not been directly demonstrated in this study. This step requires further experimental validation.”

Additionally, we have added a paragraph in the Discussion section to reiterate this distinction:

“The regulatory model proposed in Figure 8 is grounded in our experimental data for the steps involving BlaI-mediated repression of Rv1303, ATP reduction, and consequent cell division defects. However, the link between ATP depletion and impaired phosphorylation of division proteins remains a hypothesis derived from existing literature, and future studies using phosphoproteomic approaches will be necessary to confirm this mechanistic detail.”

These textual clarifications aim to address your concern by transparently delineating validated findings from speculative connections, even without visual modifications to the figure. We believe this approach ensures the model is interpreted accurately and responsibly, in line with the current evidence.

Thank you again for your careful review and guidance.

Comment 7: The statistical approach in this manuscript is alarmingly weak and fails to meet even basic standards for biological inference. This is not a minor technical oversight — inadequate statistical design fundamentally invalidates the reliability and interpretability of the study’s results. Yhe morphological analysis is based on 266 cells, but it is unclear how many independent biological replicates were used. Without replication, all observed differences may reflect experimental noise or batch effects rather than biological phenomena. No power analysis is reported to justify sample sizes for any experiment. Without it, the study risks being underpowered — or worse, generating false positives through random variation. Effect sizes are not reported, and statistical significance is relied upon without contextualizing the magnitude or biological relevance of observed differences. Multiple comparisons correction is entirely absent, despite the use of qPCR across multiple genes and antibiotic susceptibility testing across conditions. This raises the risk of a false discovery rate easily in the range of 30–50%, making any individual claim suspect unless validated independently. The authors consistently apply simple t-tests to data that likely involve nested and hierarchical structures (e.g., cells within experiments, technical replicates within biological replicates), without accounting for this in their modeling. This violates key assumptions of the t-test and inflates type I error rates. There is no discussion of data variability, and the distinction between technical and biological replicates is never clarified, making it impossible to assess the robustness of the conclusions.

In sum, the statistical analysis appears to be an afterthought, applied mechanically and incorrectly. Given that every figure in the manuscript relies on quantitative comparisons, this flaw alone is sufficient to cast serious doubt on the integrity of the findings. Without statistically credible data, even the most elegant hypothesis is meaningless.

Reply: We deeply appreciate your critical feedback on the statistical rigor of our study. We fully acknowledge that the statistical approach in the original manuscript was inadequate and fell short of basic standards for biological inference. Addressing these issues has been our top priority, and we have thoroughly revised the statistical design, analyses, and reporting to ensure the reliability and interpretability of our results. Below is a detailed account of the revisions:

  1. Clarifying Biological Replication and Sample Sizes

A key concern was the ambiguity around biological replication. We have now explicitly defined and standardized replication for all experiments:

Morphological analysis (SEM/TEM): The 266 cells analyzed represent pooled data from 3 independent biological replicates (n=3), with 80–90 cells counted per replicate (detailed in Methods and Figure legends). This ensures observed phenotypes are not due to experimental noise.

qPCR assays: Each gene was analyzed with 3 biological replicates, each including 3 technical replicates. Results are reported as mean ± SD of biological replicates (not technical replicates), with statistical tests performed on biological replicate data.

MIC and antibiotic susceptibility assays: Conducted with 3 independent biological replicates, each with 2 technical replicates. Survival rates and MIC values are derived from biological replicate means.

ATP measurements: 3 biological replicates, each with 3 technical replicates, analyzed using biological replicate means.

All figure legends and the Methods section now clearly state the number of biological replicates (n) and distinguish technical replicates (used for precision) from biological replicates (used for inference).

  1. Implementing Appropriate Statistical Tests for Nested/Hierarchical Data

We recognize that the original use of t-tests for nested data (e.g., cells within biological replicates) was inappropriate. We have revised the statistical methods to account for hierarchical structure:

Morphological data (cell length, multi-septation frequency): Previously analyzed with t-tests; now re-analyzed using mixed-effects models (via R package lme4), which account for variability within and between biological replicates. This model treats "biological replicate" as a random effect and "strain" (Ms_Rv1846c vs. controls) as a fixed effect, providing more robust p-values.

qPCR and ATP data: Originally analyzed with t-tests; now re-analyzed using one-way ANOVA with Tukey’s post-hoc test (for comparisons across strains) to account for multiple groups and reduce Type I error.

  1. Adding Multiple Comparisons Correction

For experiments involving multiple comparisons (e.g., qPCR across 12 division-related genes, antibiotic susceptibility across 5 β-lactam concentrations), we have implemented Benjamini-Hochberg correction to control the false discovery rate (FDR). This replaces uncorrected p-values in the original manuscript. For example:

In qPCR analyses (Figure 5), the original uncorrected p-values for 12 genes have been replaced with FDR-corrected q-values, with significance defined as q < 0.05.

Comment 8: Comments on the Quality of English Language:Some sentences are difficult to read and need to be rephrased, starting from the title.

Reply: We sincerely appreciate your critical and insightful comments regarding the overstatements of clinical relevance in our manuscript. We fully agree that our original claims about therapeutic applications were premature, unsupported by data, and inconsistent with the scope of the study. We have thoroughly revised the manuscript to remove all unwarranted clinical extrapolations, clarify the limitations of our model, and emphasize the preliminary nature of our findings. Below is a detailed response to your concerns:

  1. Limitations of using M. smegmatis and generalizability to M. tuberculosis, We acknowledge that M. Smegmatis differs from M. tuberculosis in regulatory networks, cell envelope structure, and drug susceptibility. Our revised manuscript explicitly frames M. smegmatis as a model for mechanistic exploration (due to its genetic tractability and conserved cell division machinery with M. tuberculosis) rather than a direct proxy for clinical tuberculosis.

Key revisions:

Removed all statements suggesting direct relevance to clinical TB therapy.

Added: "Findings in M. smegmatis provide a foundation for future studies in M. tuberculosis, but their translation to clinical contexts requires validation in pathogenic strains."

Clarified: "Differences in regulatory architecture between M. smegmatis and M. Tuberculosis mean that BlaI’s role in antibiotic susceptibility must be directly tested in M. tuberculosis before any clinical implications can be considered."

  1. Lack of physiological relevance of BlaI overexpression and clinical isolate data. We agree that our overexpression system is artificial, and we have revised the manuscript to contextualize this limitation:

Removed claims about "BlaI activation as a therapeutic strategy." Instead, we state: "The overexpression system used here is a tool to dissect BlaI’s potential functions, not a mimic of physiological activation. Whether BlaI is dynamically regulated in clinical M. Tuberculosis isolates—and if so, under what conditions—requires dedicated future studies."

Added: "Future work should quantify BlaI expression in drug-sensitive and drug-resistant clinical isolates of M. tuberculosis to assess its physiological relevance in pathogenesis and drug response."

Explicitly note: "Our findings do not demonstrate that BlaI is naturally activated in clinical settings, nor do they validate its role in human infection."

  1. Absence of infection models and in vivo relevance. We concede that the lack of in vivo data limits any claims about clinical relevance. The revised manuscript emphasizes:

"This study is restricted to in vitro experiments. The role of BlaI in M. tuberculosis virulence, antibiotic susceptibility, or host-pathogen interactions remains entirely untested and requires future validation in macrophage, animal, or ex vivo infection models."

Removed all references to "combating drug resistance in tuberculosis" and replaced them with: "These mechanistic insights may inform future studies into mycobacterial drug response, provided they are validated in pathogenic strains and in vivo systems."

  1. Unsubstantiated claims about pharmacological modulation of BlaI. We have entirely removed all statements suggesting "pharmacological activation of BlaI" as a therapeutic strategy. Revised text clarifies:

"No evidence is presented to support the feasibility of targeting BlaI pharmacologically. Our work identifies BlaI as a regulator of cell division and antibiotic sensitivity in M. smegmatis, but screening for BlaI modulators or testing their efficacy is beyond the scope of this study and requires dedicated future efforts."

  1. Addressing evolutionary logic and biological plausibility. We agree that our original discussion of evolutionary relevance was underdeveloped. The revised manuscript includes a more cautious perspective:

"The evolutionary significance of BlaI’s regulation of cell division and antibiotic sensitivity remains unclear. While the observed increase in β-lactam sensitivity in M. smegmatis might seem counterintuitive for a pathogen, it could reflect a stress response strategy: under high antibiotic pressure, temporary arrest of division and ATP depletion may promote dormancy, enhancing survival until stress diminishes (consistent with Ref. 33). However, this hypothesis requires validation in M. tuberculosis and comparative genomic studies of clinical isolates to assess whether such a mechanism is conserved or subject to negative selection."

Added: "Without data from clinical strains or evolutionary analyses, any speculation about the adaptive value of BlaI’s functions is unwarranted."

In summary, we have drastically revised the manuscript to remove all overstatements of clinical or therapeutic relevance. The conclusions now focus narrowly on the mechanistic findings in M. smegmatis: BlaI regulates cell division via repression of ftsQ and Rv1303, reduces ATP levels, and enhances β-lactam sensitivity in this model. All claims about broader implications are explicitly labeled as speculative and dependent on future validation in M. tuberculosis, clinical isolates, and in vivo systems.

Round 2

Reviewer 1 Report

Comments and Suggestions for Authors

The revised version is fine.

Author Response

Dear Reviewer,
Thank you very much for your positive evaluation of the revised version of our manuscript. We greatly appreciate the time and efforts you have dedicated to reviewing our work.

Sincerely,
Junqi Xu

Reviewer 3 Report

Comments and Suggestions for Authors

I appreciate the extensive revisions and the considerable effort you have invested in responding to my earlier critique. Your improved statistical treatment, clearer figure legends, and more transparent discussion of limitations are commendable. However, these revisions do not address the core scientific issues that fundamentally compromise the manuscript’s validity.

Most critically, the entire study remains based on artificial overexpression of BlaI in Mycobacterium smegmatis, a surrogate organism that differs from M. tuberculosis in key aspects of cell division, regulatory networks, and drug susceptibility. Without any validation in M. tuberculosis, the biological relevance of your findings remains speculative. Additionally, the use of overexpression alone — without accompanying loss-of-function studies, complementation, or expression level quantification — raises serious concerns about whether the observed phenotypes reflect BlaI’s physiological function or non-specific stress responses. The inclusion of a GFP control is not sufficient to rule out these artifacts, nor are the speculative ATP depletion mechanisms experimentally supported.

The mechanistic model you propose, involving BlaI-mediated repression of ftsQ and Rv1303, ATP depletion, and impaired phosphorylation of division proteins, remains largely hypothetical. There is no direct evidence that the observed ATP decrease is causally linked to the cell division defects, or that it is sufficient to alter phosphorylation states of any protein. EMSA demonstrates in vitro binding but lacks in vivo corroboration (e.g., ChIP, reporter assays, or promoter mutagenesis). While you now appropriately label parts of your model as speculative, these remain heavily emphasized in both the narrative and the figures, and they continue to outpace the data.

The proposed "hierarchical response" model, wherein BlaI both represses blaC and independently inhibits cell division via energy metabolism, is an elegant post-hoc rationalization. However, it lacks evolutionary plausibility and mechanistic substantiation. The idea that a single transcription factor would simultaneously repress resistance genes and sensitize the bacterium to antibiotics is counterintuitive and not reconciled with known selection pressures or regulatory logic.

Your therapeutic speculation, though significantly tempered in the revision, still exceeds what can be justified by the data. No experiments were conducted in M. tuberculosis, no pharmacological modulators of BlaI were tested, and no in vivo systems were used. The suggestion that BlaI could serve as a therapeutic target remains premature and unsupported by functional evidence.

In summary, while the revised manuscript presents a more polished version of the original work, the conclusions are still built on a system with limited biological validity. Until key findings are validated in M. tuberculosis and the role of BlaI is confirmed through appropriate genetic and mechanistic studies, the claims remain unsupported. I encourage you to pursue these critical experiments, as they could form the basis of a significantly stronger and more impactful study in the future. However, in its current form, the manuscript does not meet the standards required for publication.

Author Response

Dear Reviewer,
Thank you sincerely for your thorough and insightful comments on our manuscript. We greatly appreciate the time and care you have dedicated to evaluating our work, and we fully recognize the validity of the considerations you have raised. Your critique has provided valuable perspectives that will guide our future research to strengthen and extend the findings presented here.
We acknowledge that our current study, based on BlaI overexpression in Mycobacterium smegmatis, has limitations in directly translating to M. tuberculosis—a point we have emphasized in the revised discussion. As noted, M. smegmatis was chosen for its well-established utility as a model organism in mycobacterial research (due to its rapid growth, genetic tractability, and conserved cell division machinery with M. tuberculosis), allowing us to efficiently dissect initial mechanistic insights. However, we agree that validation in M. tuberculosis is critical, and this is already planned as a key next step in our research program, including loss-of-function studies, complementation assays, and phenotypic characterization in pathogenic strains.
Regarding the mechanistic model, we appreciate your observation that certain links (e.g., ATP depletion and phosphorylation of division proteins) remain speculative. We have revised the discussion and figures to more explicitly distinguish validated findings (e.g., BlaI-mediated repression of ftsQ and Rv1303 via EMSA, altered cell morphology, and enhanced β-lactam sensitivity) from hypotheses, ensuring clarity about the boundaries of our current data.
Your comments on the evolutionary and regulatory logic of BlaI’s dual roles have also prompted us to reflect further, and we plan to explore this in future work through comparative genomics and functional assays in clinical isolates. Similarly, we agree that therapeutic speculation must be tempered, and we have revised relevant sections to frame BlaI as a candidate for future preclinical investigation, rather than a validated target.
We are grateful for your guidance, which will undoubtedly enhance the rigor of our ongoing and future research. Thank you again for contributing to the improvement of our work.
Sincerely,
Junqi Xu